



# Description and performance of the CARMA sectional aerosol microphysical model in CESM2

Simone Tilmes[1], Michael J. Mills[1], Yunqian Zhu[2,3], Charles G. Bardeen[1], Francis Vitt[1], Pengfei Yu[4], David Fillmore[1], Xiaohong Liu[5], Brian Toon[3,6], and Terry Deshler[6,7]

[1]Atmospheric Chemistry Observations and Modeling Laboratory, National Center for Atmospheric Research, Boulder, CO, USA

[2]Earth System Research Laboratory, National Oceanic and Atmospheric Administration, Boulder, CO, USA.

[3]Department of Atmospheric and Oceanic Sciences, University of Colorado, Boulder, CO, USA.

[4]Institute for Environment and Climate Research, Jinan University, Guangzhou, China.

[5]Department of Atmospheric Sciences, Texas A&M University, College Station, TX, USA.

[6]Laboratory for Atmospheric and Space Physics, University of Colorado, Boulder, CO, USA.

[7]Previously at the Department of Atmospheric Science, University of Wyoming, Laramie, WY, USA.

**Correspondence:** Simone Tilmes (tilmes@ucar.edu)

**Abstract.** We implemented the Community Aerosol and Radiation Model for Atmospheres (CARMA) in both the high and low-top model versions of the Community Earth System Model Version 2 (CESM2). CARMA is a sectional microphysical model, which we use for aerosol in both the troposphere and stratosphere. CARMA is fully coupled to chemistry, clouds, radiation, and transport routines in CESM2. This development enables the comparison of simulations with a sectional (CARMA) and a modal (MAM4) aerosol microphysical model in the same modeling framework. The new implementation of CARMA has been adopted from previous work with some additions that align with the current CESM2 MAM4 implementation. The main updates include an interactive secondary organic aerosol description in CARMA using the volatility basis set (VBS) approach, updated wet removal, and the use of transient emissions of aerosols and trace gases. In addition, we implemented an alternative aerosol nucleation scheme in CARMA, which is also used in MAM4. Detailed comparisons of stratospheric aerosol properties after the Mt Pinatubo eruption reveal the importance of prescribing sulfur injections in a larger region rather than in a single column to better represent the observed evolution of aerosols. Both CARMA and MAM4 in CESM2 are able to represent stratospheric and tropospheric aerosol properties reasonably well compared to observations. Several differences in the performance of the two aerosol models show, in general, an improved representation of aerosols using the sectional aerosol model in CESM2. These include a better representation of the aerosol size distribution after the Mt Pinatubo volcanic eruption in CARMA compared to MAM4. MAM4 produces on average smaller aerosols and less removal than CARMA, which results in a larger total mass. Both CARMA and MAM4 reproduce stratospheric Aerosol Optical Depth (AOD) within the errorbar of the observations between 2001 and 2020, except for recent larger volcanic eruptions that are overestimated by both model configurations. The CARMA background surface area density and aerosol size distribution in the stratosphere and troposphere compare well to observations, with some underestimation of the Aitken-mode size range. MAM4 shows shortcomings in reproducing coarse-mode aerosol distributions in the stratosphere and troposphere. This work outlines additional development needs for CESM2 CARMA to improve the model compared to observations in both the troposphere and stratosphere.





# 1 Introduction

Earth System Models (ESMs) are necessary tools to understand the effects of natural and anthropogenic influences on the climate system in the past and present and are essential to predict future changes. These models parameterize complex interactions between different Earth system components to be efficient enough to run on current supercomputer systems with reasonable throughput. A range of parameterizations with different complexity has been developed to reproduce physical processes reasonably well for specific scientific applications. To run long climate simulations, simplified schemes for chemistry and aerosols have been developed that perform well compared to observations (Danabasoglu et al., 2020). However, simplified schemes lack physical interactions, such as the coupling between aerosol and chemistry in the stratosphere, as included in a more comprehensive model configuration (Mills et al., 2016, 2017). More sophisticated parameterizations are necessary to understand the possible shortcomings in simplified parameterizations and to reduce uncertainties in ESMs predictions.

Here, we focus on the representation of aerosols in the troposphere and stratosphere. Aerosols play an important role in both climate (Kremser et al., 2016) and air quality (e.g., Fiore et al., 2015). Large uncertainties exist in aerosol formation, cloud and aerosol coupling, effects on radiation and chemistry, and removal of aerosols. Different aerosol schemes have been developed in ESMs, reaching from simplified bulk aerosol models with fixed sizes and externally mixed aerosols (e.g., Chin et al., 2002; Colarco et al., 2010) to modal representations of the aerosol distribution assuming internally mixed aerosols within each mode (modal aerosol models) (e.g., Liu et al., 2012), to the most complicated size-resolved representation of the atmospheric aerosol distributions also called sectional aerosol models (e.g., Kokkola et al., 2018; Sukhodolov et al., 2021). Depending on these representations, interactions between aerosols and other components (clouds, chemistry, and radiation) need to be adjusted to accommodate the specifics of the aerosol scheme.

The purpose of this work is to describe and evaluate the performance of a sectional aerosol model for both troposphere and stratosphere, following the implementation by Yu et al. (2015), into different atmospheric configurations of the Community Earth System Model Version 2 (CESM2). The sectional aerosol model used here is a configuration of the Community Aerosol and Radiation Model for Atmospheres (CARMA). CARMA is a framework for sectional and also referred to as size-resolved cloud and aerosol models (Toon et al., 1988; Bardeen et al., 2008, 2013; Yu et al., 2015; Zhu et al., 2015, 2017; Yu et al., 2022). The CARMA aerosol model has been previously coupled to CESM Version 1 using the Community Atmospheric Model Versions 4 and 5 (CAM4 and CAM5) with tropospheric and stratospheric chemistry and resulted in improved aerosol representation compared to a modal aerosol model based on various comparisons with observations (e.g., Yu et al., 2015, 2016, 2017; Murphy et al., 2021). In this study, we compare the two different aerosol models (CARMA and MAM4) using two CESM2 atmospheric configurations, CAM6 with comprehensive tropospheric and stratospheric chemistry (CAM6chem) and the Whole Atmosphere Community Climate Model Version 6 with middle atmospheric chemistry (WACCM6-MA). These configurations include the coupling to chemistry, radiation, optics, cloud-aerosol interactions, emissions, and wet and dry removal.

The new implementation in CESM2, as discussed here, allows running the two available aerosol models (MAM4 and CARMA) within the same code base. Simulations with the same dynamical core, radiation scheme, chemistry, and trans-



port scheme and with nudged meteorological fields, e.g., winds and temperatures, are performed to identify differences that are, for the most part, based on the aerosol scheme and related couplings. Some improvements to the Yu et al. (2015) CARMA aerosol model and the atmospheric coupling have been implemented to align it with some recent atmospheric model developments. These include updates in the wet removal scheme and the description of secondary organic aerosols (see Section

2.2). In addition, CARMA coupled to high-top model WACCM6 (Gettelman et al., 2019; Davis et al., 2023) allows improved representation of stratospheric transport and dynamics compared to the low-top model. In contrast to an earlier CARMA version coupled to WACCM Version 4 (English et al., 2012), aerosols are radiatively active. In this paper, we evaluate aerosols and optical properties in the stratosphere and troposphere, including the impacts of small and large volcanic eruptions and the aerosol background composition based on in-situ and satellite observations. Effects on chemistry are only briefly evaluated.

The implementation of the optional sectional aerosol model CARMA in the atmospheric model of CESM2 and its evaluation is the first step towards a fully coupled CESM2 CARMA configuration, including ocean, and sea ice, which will allow fully coupled climate simulations.

The paper is organized as follows: Details of the model and the two aerosol microphysical schemes used are given in Section 2. This also includes details of the coupling between CESM2 and CARMA or MAM4 with regard to various processes

covering cloud-aerosol interactions, dry and wet removal, radiation and optics, chemistry, and emissions. We further outline the computational performance of the different configurations used in this work. Section 3 describes the experimental design of the work. Results on stratospheric aerosol performance are summarized in Section 4, with details on the performance of the model to simulate the aerosol evolution after the Mt Pinatubo volcanic eruption based on sensitivity tests. Background stratospheric aerosol properties and ozone are also evaluated. Section 5 focusses on tropospheric aerosol model performance

between 2001-2020 and between 2016 and 2018 compared to the NASA ATom aircraft mission. We close with a discussion and suggestions for further model development in Section 6 and conclude thereafter.

## 2 Model description

### 2.1 CESM2.2 model configurations

Experiments performed in this study are based on two different atmospheric configurations, CAMchem and WACCM-MA of

the Community Earth System Model (CESM2.2) (Danabasoglu et al., 2020). CAMchem includes comprehensive chemistry in the troposphere and stratospheric (TS1) (Emmons et al., 2020) with some minor updates added in this study and uses a configuration with $0.9°x1.25°$ degree in the horizontal resolution, 32 levels in the vertical with a top at around 42 km. The aerosol model includes a volatility basis set (VBS) secondary organic aerosol scheme (Tilmes et al., 2019), including interactive biogenic emissions from the Model of Emissions of Gases and Aerosols from Nature version 2.1 MEGAN2.0 (Guenther et al.,

2012). This model version is frequently used for air quality studies in the troposphere (e.g., Gaubert et al., 2021; Tang et al., 2022), and for studies in the Upper Troposphere and Lower Stratosphere (UTLS). It also performs well compared to observations in the stratosphere (Emmons et al., 2020).





WACCM-MA is a high-top version of CESM and has 70 vertical levels with a model top at about 150 km. It has been designed for studies that focus on stratospheric chemistry and circulation, including impacts of volcanic eruptions and strato-
spheric aerosol injection. For example, the first Geoengineering Large Ensemble Simulations (GLENS) (Tilmes et al., 2018) used an earlier WACCM-MA version with 0.9° x 1.25° horizontal resolution. In this work, we use CESM2(WACCM-MA) with 1.9° x 2.5° horizontal resolution (Davis et al., 2023). This model version, coupled with a full ocean, shows a reasonable climate response, and its dynamics and chemistry in the stratosphere are comparable to the 0.9° x 1.25° WACCM6 version with comprehensive tropospheric and stratospheric chemistry. The model includes comprehensive chemistry in the stratosphere,
mesosphere, and lower thermosphere but only represents chemistry with limited complexity in the troposphere (Davis et al., 2023). In turn, secondary organic aerosols are only represented in a simplified manner.

## 2.2   Standard Aerosol description in CESM2 using the Modal Aerosol Model (MAM4)

The default CESM2 aerosol scheme in CAMchem and WACCM is the modal aerosol model (MAM4) (Liu et al., 2012, 2016), with updated prognostic stratospheric sulfate aerosols (Mills et al., 2016). MAM4 microphysics describes four modes: the
Aitken, accumulation, coarse and primary carbon modes. The primary carbon mode has been added to represent the aging processes of black carbon and primary organic matter while being coated by soluble species (sulfate and organics) with monolayers (Liu et al., 2016). The geometric standard deviation in MAM4 for the different modes is Aitken: 1.6, accumulation: 1.6, and coarse:1.2. (Liu et al., 2016; Mills et al., 2016). Table 1 describes the model settings of the size range, particle types, and morphology for MAM4 and CARMA aerosols.
The microphysics of MAM4 include a binary parameterization (Vehkamäki et al., 2002) for the sulfuric acid vapor ($H_2SO_4$-$H_2O$) homogeneous nucleation for new particle formation. The loss of the new particles by coagulation, as they grow from critical cluster size to Aitken mode size, is accounted for using the parameterization by Kerminen and Kulmala (2002). The condensation of $H_2SO_4$ vapor is treated dynamically using a standard mass transfer expression that is integrated over the size distribution of each mode (Binkowski and Shankar, 1995). An accommodation coefficient of 0.65 is used for $H_2SO_4$
and other species (Pöschl et al., 1998). In the troposphere, $H_2SO_4$ condensation is treated as irreversible, while SOA (gas) condensation is reversible and based on equilibrium vapor pressure over particles. Evaporation of sulfate particles is included only above the tropopause (Mills et al., 2016). Coagulation of the Aitken, accumulation, and primary carbon modes is treated within each and between different modes. It reduces the number but leaves mass unchanged. For tropospheric aerosol, water uptake in MAM4 is based on the equilibrium Köhler theory (Ghan and Zaveri, 2007) using the relative humidity and the
volume mean hygroscopicity for each mode to diagnose the wet volume mean radius of the mode from the dry volume mean radius. Gravitational settling velocities are calculated as a function of altitude (Seinfeld and Pandis, 1998). For the stratosphere, sulfates are in equilibrium with the water, and a weight percent of $H_2SO_4$ is calculated based on the parameterization by Tabazadeh et al. (1997). Settling velocities depend on wet particle size and mass and are, therefore, different between modes.





### 2.3 Community Aerosol and Radiation Model for Atmospheres (CARMA)

The CARMA aerosol model (version 4.3) for the troposphere and stratosphere (denoted to just "CARMA" in the following) includes prognostic aerosols in both troposphere and stratosphere, as described in detail in Yu et al. (2015). The implementation is further based on previous aerosol descriptions for sea-salt (Fan and Toon, 2011), dust storms (Su and Toon, 2009), and stratospheric sulfates (English et al., 2013). Additional implementations, such as the inclusion of volcanic ash (Zhu et al., 2020), new descriptions of polar stratospheric clouds (PSCs) (Zhu et al., 2015, 2017), polar mesospheric clouds (PMC) (Bardeen et al.,

2010), and sectional nitrate and ammonium (Yu et al., 2022), are not included in the current version of the model but will be included in future work.

CARMA can be configured with numerous classes of particles or groups. We employ an internally mixed group composed of primary and secondary organics, black carbon, sulfate, dust, and sea salt, and a pure sulfate group that only includes sulfates (see Table 1). The pure sulfate group includes nucleation of $H_2SO_4$, condensation, and coagulation with both the pure and

mixed groups. The pure sulfate group can be used to identify geographic regions of active nucleation. CARMA keeps track of the total mass and the core masses (or elements) of each group in each mass bin, with sulfuric acid as the volatile component of each bin. Currently, CARMA only allows one component of a group to be volatile. The addition of SOA in this model requires calculating SOA volatility (gas to aerosol exchange) in CAM. Each group is described as individual discrete aerosol mass bins. Here, we use 20 mass bins as defined in Yu et al. (2015). The bins track the dry mass of the particles and assume water is in

equilibrium to calculate the wet radius of the particle. The mixed aerosol group defines these bins between 0.05 –8.7 $\mu$m in radius and the pure sulfate group from 0.2 nm to 1.3 $\mu$m. In addition, CARMA is capable of resolving many more arbitrary distributions of aerosol sizes, in contrast to the minimalist approach of MAM4, which assumes a superposition of only four lognormal modes (with two of those, the primary carbon mode and the accumulation mode covering very similar size ranges).

Microphysical processes in CARMA include binary homogeneous nucleation of sulfuric acid and water (Zhao and Turco,

1995) (called "Zhao scheme" in the following) and sulfuric acid evaporation (Toon et al., 1989) for the pure sulfate group only; sulfuric acid condensation, and gravitational settling for both groups; and aerosol coagulation within and between the mixed and pure groups, including the effects of Vander Waals forces (English et al., 2011). In addition to the Zhao scheme, in this work, we added the binary homogeneous nucleation scheme, described in Vehkamäki et al. (2002) (called "Vehkamäki scheme" in the following). This nucleation scheme is also used in MAM4, as the default. Vehkamäki et al. (2002) employ

an improved model for hydrate formation valid for both tropospheric and stratospheric conditions and uses a parameterization based on observations. In contrast, the Zhao scheme is based on a physical approach and was developed and validated primarily for stratospheric conditions. The effects of the two schemes are compared for the Mt Pinatubo eruption in 1991 (Section 4.1.2.) and for tropospheric background conditions (Section 5.2.4.). The model does not currently employ nucleation influenced by ammonia or organics, which is likely important near the ground.

For the pure sulfate group, as for MAM4 in the stratosphere, the wet radius of the particle is determined by the weight percent of $H_2SO_4$ in the $H_2SO_4/H_2O$ particles, based on the parameterization by Tabazadeh et al. (1997). For the mixed radius, the wet radius is parameterized based on the relative humidity and the weighted hygroscopicity considering the composition of the



**Table 1.** Aerosol specifics for CARMA and MAM aerosol microphysical models coupled to WACCM-MA and CAMchem. Species names used here are specific to each aerosol model.

| Aerosol Model | CARMA | MAM4 |
|---|---|---|
| Size Description | 40 bins (20 per group)<br>Mixed Group: 0.05–8.7 $\mu$m<br>Pure Group: 0.2 nm to 1.3 $\mu$m | Primary Carbon (0.06–0.30$\mu$m)<br>Aitken (0.015–0.053$\mu$m)<br>Accumulation (0.058–0.48$\mu$m)<br>Coarse Modes (0.4-40$\mu$m) |
| Species types | sulfate, p-organic, s-organic,<br>black-carbon, sea-salt, dust | sulfate, p-organic, s-organic,<br>black-carbon, sea-salt, dust |
| Groups and species | Mixed Group: MX, Pure Group: PRSULF<br>MX: total (incl SULF), BC,OC,SALT,DUST<br>SOA (or SOA1, SOA2, SOA3, SOA4, SOA5) | Internally Mixed Modes<br>so4, pom, bc, ncl, dst<br>soa (or soa1, soa2, soa3, soa4, soa5) |
| Morphology (core/shell)<br>for optics | core: BC, DUST<br>shell: SULF, OC, SALT, H2O | |

internally mixed particles (Petters and Kreidenweis, 2007). To avoid generating too large particles through swelling, relative humidity is constrained to be less than 99.5% in CARMA when calculating the wet radius and wet density of particles. While Yu et al. (2015) assumed no particle swelling below 190 K, in this study, we use the relative humidity at 190 K to calculate the particle swelling. Therefore the wet radius below 190 K. CARMA further includes parameterization of emissions of sea salt and dust, as well as the removal of aerosols through wet and dry deposition that can be independent of the atmospheric model configuration, as described in Section 2.4.5.

## 2.4 Coupling between CESM2 and CARMA / MAM4

Aerosols interact with various processes in the atmosphere and need to be coupled to those components of the atmospheric model independent of the aerosol scheme. These processes include radiation and optics, chemistry-aerosol interactions, cloud-aerosol interactions, emissions, and wet and dry deposition, as illustrated in Figure 1. MAM4 aerosol microphysical processes are integrated into the workflow of the CESM2 atmospheric model. In contrast, CARMA has been integrated as a stand-alone model, resulting in a slightly different ordering than MAM4. The order of applied physical processes is indicated as numbers black for MAM4 and red for CARMA in Figure 1.

Physical processes within The Community Atmospheric Model (CAM) are time split, meaning that processes happen sequentially in a specified order rather than all at once (Williamson, 2002). Physical processes are divided between processes that occur before coupling and after coupling with surface processes (e.g., land, ocean). The processes before coupling besides advection and convection (not included in Figure 1) include deep convection Zhang and McFarlanle (1995), planetary

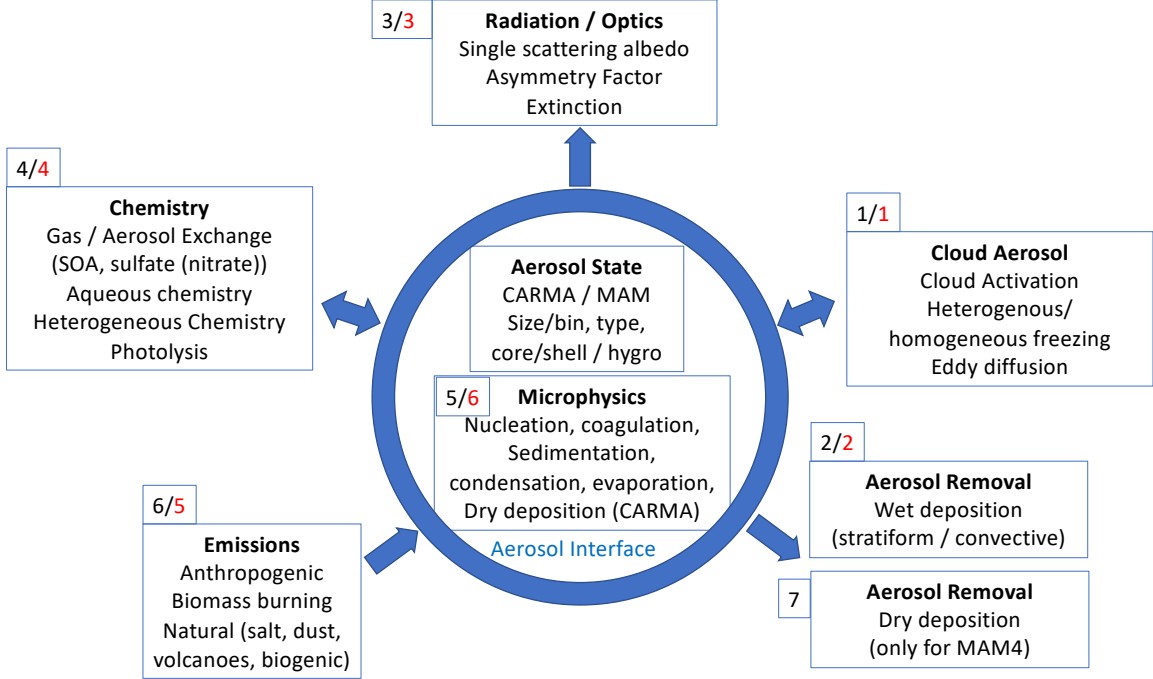

**Figure 1.** Schematic of the coupling between different Aerosol processes and CAMchem and WACCM-MA. The order of processes related to aerosols in CESM2-CAM6 is illustrated in small numbers (black for MAM4 and red for CARMA) given at each process box. The blue circle separates the processes that occur as part of the aerosol microphysical scheme from the processes that are specific to the atmospheric model (CAM6).

boundary layer, shallow convection, and moist turbulence (Bogenschutz et al., 2012). These are coupled to aerosol activation (Abdul-Razzak and Ghan, 2000), eddy diffusion (Process 1 in Figure 1), two-moment cloud microphysics (**?**), and convective and stratiform wet removal (Process 2 in Figure 1). After that, optical properties and radiative transfer are calculated (Process 3 in Figure 1), followed by the coupling to the land and ocean.

In CAM6, surface emission fluxes for gases and aerosol, including anthropogenic, biomass burning, biogenic, and ocean

emissions, are calculated after the surface coupling. However, they are added to the lower atmospheric layer after chemistry. Chemistry (Process 4) includes aqueous-phase chemistry and gas-aerosol-exchange and applies vertical emissions of gases and aerosol. Chemistry also includes the MAM4 microphysical processes (Process 5 for MAM4). After chemistry, the emissions are applied (Process 6), followed by the dry removal of gases, including aerosol precursors like sulfur and VOCs (not shown in Figure 1). Finally, dry deposition of aerosol is applied (Process 7 for MAM4). For CARMA, microphysical processes are

not included in chemistry but are applied later. In this case, emissions of gases and aerosols (Process 5 for CARMA) and dry deposition of gases (not shown in Figure 1) are applied after chemistry. Aerosol microphysics for CARMA (Process 6) is done last, which includes sedimentation, dry deposition, molecular diffusion, and coagulation, followed by nucleation, growth, and evaporation, which may be sub-stepped for stability due to potentially large process rates.



### 2.4.1 Cloud-aerosol processes

Advected aerosols in the atmosphere (for modal and sectional models in CAM6) are called interstitial aerosols. The aerosols that have been activated to serve as condensation nuclei and form clouds are removed from the interstitial aerosols and are classified as the so-called cloud-borne aerosols (Easter et al., 2004). Cloud-born aerosols in CAM6 (both MAM4 and CARMA) are not advected. The transition between interstitial and cloud-borne aerosols and vice versa depends on the atmospheric conditions, including ice and liquid cloud fraction, relative humidity, and temperature.

Activation of clouds is calculated for both MAM4 and CARMA based on the critical supersaturation of air masses, which is obtained from the turbulent vertical velocity in the updraft of air masses (Abdul-Razzak and Ghan, 2000). The turbulent vertical velocity is based on subgrid processes and is currently parameterized and represented through a probability distribution function. In addition to the aerosol activation, diffusive mixing of aerosols across vertical levels has been considered for both MAM4 and CARMA. Shrinking or removing clouds leads to the evaporation of cloud-borne aerosols in the model, which

moves them back into interstitial. In MAM4 this method is applied for each log-normal mode and each species, and in CARMA it is applied for all bins and species per bin. Both MAM4 and CARMA keep track of cloud-borne particles for each mode and bin they originated from when activated. After the evaporation of clouds, aerosols are moved back into the mode or bin of their origin.

Both homogeneous and heterogeneous nucleation is considered in MAM4 and CARMA for ice crystal nucleation in mixed-

phase and cirrus clouds. It is based on the number of dust and sulfate within the mixed and the pure sulfate group in CARMA, considering only aerosols that are > 0.1 microns. For MAM4, the Aitken mode of sulfate and coarse mode of dust are considered for ice nucleation in ice clouds (Liu and Penner, 2005; Liu et al., 2007). For CARMA, nucleation, condensation, and deposition in mixed clouds < -37°C are done based on Mayers et al. (1992). In contrast, MAM4 describes heterogeneous nucleation in mixed-phase clouds based on the classical nucleation theory described in Wang et al. (2014).

### 2.4.2 Dry and wet removal of aerosols

The wet removal of aerosols, including in-cloud and below-cloud wet removal, is done by coupling to the atmospheric model (CAM6). In CAM6, in-cloud removal in shallow convective and stratiform clouds is treated seamlessly based on the cloud and precipitation information from the two-moment Morrison and Gettelman microphysics (Gettelman and Morrison, 2015). For the wet removal in deep convective clouds, CAM6 uses the Zhang and McFarlanle (1995) deep convection scheme, cou-

pled with a unified scheme for aerosol convective transport and wet scavenging by Wang et al. (2013) with updates and improvements by Shan et al. (2021). For CARMA, we also adopt the convective wet removal scheme by Wang et al. (2013) and Shan et al. (2021). An updated version of CESM1-CARMA adopted a different convective removal scheme introduced by Yu et al. (2019), which also considers the secondary activation of aerosols from entrained air above the cloud base.

Aerosol dry deposition velocities in MAM4 and CARMA are calculated using the Zhang (2001) parameterization with pre-

scribed land-use and surface layer information. Aerosol mixing ratio changes and fluxes from dry deposition and sedimentation are calculated throughout a vertical column. Differences in dry deposition fluxes between MAM4 and CARMA (see below)





are due to the differences in the particle size of the mixed group, which results in larger particles and faster sedimentation for CARMA compared to MAM4, as also discussed in Yu et al. (2015).

### 2.4.3  Radiative transfer and optics

CAM6 uses the Rapid Radiative Transfer Model for GCMs (RRTMG) (Iacono et al., 2008) for the radiative transfer calculation in longwave (16 bands) and shortwave (14 bands) including heating rates and radiative fluxes. Besides using the information on cloud fraction from liquid, ice, and snow, it requires information on aerosol extinction, single-scattering albedo, and asymmetry parameter, per wavelength band in the short wave, and absorption in both long and shortwave bands. For CARMA, the integration of optics for a core-shell representation that has been included for the mixed particle by Yu et al. (2015) is adopted

here, using lookup tables that include pre-calculated aerosol radiative properties based on the Mie theory and following the core-shell assumption by Toon and Ackerman (1981). Black carbon and dust are assumed to form the core of the mixed particles, while the other water-soluble constituents form the shell. Here, we expand the radiative properties to consider secondary organic aerosols using the lookup tables derived for organic aerosols (Yu et al., 2015). MAM4 uses the parameterization by Ghan and Zaveri (2007) that assumes an internal mixture of hydrated aerosol components with lognormal size distributions to

calculate optical properties using the wet surface mode radius. As for CARMA, pre-calculated aerosol properties based on the Mie theory are provided through lookup tables.

### 2.4.4  Coupling of aerosols to CESM2 chemistry

CAMchem and WACCM-MA include interactive chemistry in the troposphere and stratosphere. WACCM-MA includes much more simplified tropospheric chemistry resulting in less ozone and other oxidants than CAMchem (Gettelman et al., 2019;

Davis et al., 2023). Oxidants (in particular OH and ozone) are important for the formation of aerosol precursors (VOCs and $SO_2$) for both SOA and sulfate. Detailed sulfur chemistry includes dimethyl sulfide (DMS), and organic carbonyl sulfide (OCS) as important precursor emissions for the troposphere and stratosphere (Mills et al., 2016).

The formation of sulfate in the troposphere through aqueous-phase chemistry is included for MAM4 and CARMA as described in Barth et al. (2000). Aqueous-phase reactions include reactions of aqueous sulfur by ozone and hydrogen peroxide to

form $SO_4$ and therefore depend on tropospheric chemistry. The produced sulfate is added into the cloud-borne aerosol MAM4 sulfate modes or CARMA bins. The reduced oxidants in WACCM-MA are expected to lead to reduced aqueous-phase production, which generally results in larger sulfate burdens since cloud-borne sulfate is removed faster than interstitial aerosols (Barth et al., 2000). In this version of the model, we updated aqueous-phase chemistry to only be active in liquid clouds. CAM6 also included reactions on ice clouds (not used here), which has not been sufficiently established in the literature.

The formation of SOA from aerosol precursors is performed differently in CAMchem and WACCM-MA. The formation of SOA in CAM-chem is based on the volatility basis set approach that defines five different SOA gas-phase and aerosol species that experience gas-to-aerosol exchange, depending on their volatility characteristics (Hodzic et al., 2016; Tilmes et al., 2019). For WACCM-MA, a simplified SOA scheme is used where a gaseous SOA precursor is directly emitted at the surface, and only one volatility bin is considered. Depending on the chemistry in the troposphere, either one (for simplified tropospheric





chemistry in WACCM-MA) or five (for CAM-chem) elements have been added to the CARMA mixed aerosol group to represent the volatility bins and therefore, the condensed phase of SOA in the bin. The additional SOA elements are fully coupled to CARMA microphysics and are part of the mixed particle (or group). The production and loss of SOA for each element are applied to the CARMA SOA aerosols. We also include SOA photolysis, assuming a reaction rate that is 0.04 times the photolysis rate of nitrogen dioxide, as discussed in Hodzic et al. (2015), and add the SOA formation from glyoxal in aqueous

aerosols (Knote et al., 2014) as also done for MAM4 in CAMchem. For WACCM-MA, only CARMA includes photolysis of SOA, while MAM4 does not.

Aerosols in both the troposphere and stratosphere further provide surfaces for heterogeneous reactions, e.g., affecting chemical reactions. In the troposphere, surface area density affecting heterogeneous reactions is calculated based on the mass and effective radius of sulfate, organic aerosols, and black carbon. For MAM4, the primary carbon mode (black carbon and primary

organic matter) is not included in heterogeneous chemistry (Tilmes et al., 2015). In the stratosphere, both MAM4 and CARMA include surface area density for six heterogeneous reactions, with varying rates for sulfate, nitric acid trihydrate, and water ice (Mills et al., 2016).

### 2.4.5 Emissions of aerosols

Surface and vertical emissions for anthropogenic, biomass burning, soil, and volcanic gases are prescribed for all experiments

(see Section 3). The oceanic fluxes of DMS are calculated using the Online Air-Sea Interface for Soluble Species (OASISS) (Jo et al., 2023). Dust and sea salt emissions are calculated as part of the aerosol model. CARMA uses size-dependent dust and sea-salt source functions described in detail in Yu et al. (2015). Briefly, the calculation of sea salt and dust emissions is based on 10-meter winds from the atmospheric model and applies a Weibull wind distribution (Gillette and Passi, 1988) to represent subgrid wind velocity. For calculating sea salt emissions, we use the sea-spray aerosol source function introduced

by Fan and Toon (2011), which combines different source functions for different aerosol size ranges. In contrast to Yu et al. (2015), marine organic aerosols are not included in CARMA, to be consistent with MAM4. We use a 1x1 degree fixed soil erodibility file to calculate dust emissions and apply a dust emission scaling factor of 0.5 for the 1-degree CAMchem version and 0.4 for WACCM MA.

MAM4 sea-salt and dust emission fluxes are described in Liu et al. (2012), with updates for the dust emission size distribu-

tions. The sea salt emissions are based on the scheme by Mårtensson et al. (2003), derived for dry diameter (Dp) < 2.8 $\mu$m and the (Monahan et al., 1986) scheme for Dp > 2.8 $\mu$m, both of which depend on the 10-meter wind with the former one also depending on ocean water temperature. The dust emissions are calculated following the scheme of Zender (2003), with the emission size distribution calculation updated to be based on Kok (2011).

### 2.5 Computational performance

The CARMA size-resolving aerosol model includes 193 additional advected aerosol tracers for CAMchem and 121 for WACCM-MA (Table 2). The increase in advected tracers in CARMA compared to the modal aerosol model configuration adds significantly to the computational costs of the atmospheric host model. Using CAMchem with 0.9° x 1.25° degree hori-





**Table 2.** Model configurations and experiments between 2001-2020

| Model configuration | CAMchem | WACCM-MA | CAMchem | WACCM-MA |
|---|---|---|---|---|
| Horizontal Resolution | 0.9x1.25 | 1.9x2.5 | 0.9x1.25 | 1.9x2.5 |
| Top of Model | 42km | 150km | 42km | 150km |
| Chemistry | TS1 | MA | TS1 | MA |
| Aerosol | CARMA | CARMA | MAM4 | MAM4 |
| Number of Aerosol Tracers | 220 | 140 | 27 | 19 |
| Throughput | 2.6 yrs/day | 2.5 yrs/day | 3.6 yrs/day | 9.2 yrs/day |
| Model Cost (Core hours/yr) | 31 K | 11 K | 7.5 K | 2.3 K |
| Nucleation Scheme | Zhao | Zhao | Vehkamäki | Vehkamäki |

zontal resolution increases the model costs from $\approx$ 7500 core hours per year (for MAM4) to $\approx$ 31,000 core hours per year of simulation for CARMA, with a somewhat smaller throughput for CARMA in the specific configuration. WACCM-MA using

1.9° x 2.5° horizontal resolution requires 11,000 core hours per year for CARMA. In comparison, the MAM4 configurations used here require 2300 core hours per year, including a much better throughput of 9.2 years per day of simulation in MAM4 compared to 2.5 years per day for CARMA.

Due to the long run times, CARMA model configurations need to be carefully chosen regarding scientific needs and model costs. With the current configurations, decade-long simulations are easily possible. For studies of aerosols in the troposphere

and UTLS, and to study the impacts of pyroCb events, the low-top CAMchem has been used successfully in the past (e.g., Yu et al., 2019). Complex tropospheric chemistry is required for tropospheric aerosol formation, which affects aerosol burden, including secondary organic aerosols. A higher horizontal resolution is desired to better simulate the effects of meteorological variability and climate impacts. Tropospheric aerosol formation and composition may also be essential to investigate stratospheric background aerosol since tropospheric aerosols and their precursors are naturally injected into the stratosphere, for

example, through the upper tropical troposphere and the Asian monsoon anticyclone. They may also matter for evaluating solar-powered lofting experiments (Gao et al., 2021). On the other hand, the WACCM-MA configuration is more suited for stratospheric-focused experiments, including investigating the effects of volcanic eruptions or stratospheric aerosol injections on stratospheric chemistry and dynamics. However, this configuration, while relatively cheap, does not produce an interactive Quasi-biennial Oscillation (QBO) and may therefore not be optimal for specific research questions. Other configurations

may be used, including WACCM-MA with a 1-degree horizontal resolution, but its current model costs are around 70 K core hours per simulated year. Even more expensive is the WACCM 1-degree model version with full tropospheric and stratospheric chemistry (not evaluated here).





**Table 3.** Model Experiments 1990-1995, using WACCM-MA

| Aerosol | CARMA | CARMA | CARMA | CARMA | MAM4 | MAM4 |
|---|---|---|---|---|---|---|
| Mt.Pinatubo injection | 5 TgS | 5 TgS | 5 TgS | 7 TgS | 5 TgS | 5 TgS |
| Injection location | 15N, 120E | 5S-15N, zonal | 5S-15N, zonal | 5S-15N, zonal | 15N, 120E | 5S-15N, zonal |
| Injection altitude | 18-20km | 19-27km | 19-27km | 19-27km | 18-20km | 19-27km |
| Nucleation Scheme | Zhao | Zhao | Vehkamäki | Vehkamäki | Vehkamäki | Vehkamäki |

## 3 Experimental Design

Two sets of model experiments are performed using different configurations of CESM2. All the model simulations use observed
sea surface temperatures and sea ice conditions and are nudged every 12 hours to winds and temperatures using MERRA2
meteorological reanalyses (Davis et al., 2022). The first set of experiments between 1990 and 1995 focuses on the period
shortly before and after the largest recent volcanic eruption of Mt Pinatubo in June 1991 (Table 3). Here, different model
experiments are compared using WACCM-MA with CARMA and MAM4, Section 2.1 (using 1.9° x 2.5° degrees (or "2deg")
horizontal resolution). The model simulations start from a historical WACCM-MA 2deg simulation with prescribed sea-surface
temperatures in 1990. For CARMA, a three-year spin-up period was added to properly build up background aerosols. The
second set of experiments focuses on the performance of stratospheric background aerosol conditions and effects of small
volcanoes between 2001 and 2020 and the performance of tropospheric aerosol properties (Table 2). WACCM-MA simulations
continued after 1995 (from the first set of experiments) for CARMA and MAM4. CAMchem configurations (using 0.9° x
1.25° degrees (or "1deg") horizontal resolution, see Section 2.1) were started using initial conditions taken from the historical
WACCM TSMLT CMIP6 simulations.

Between 1990 and 2000, we use CMIP6 emissions for anthropogenic, biomass burning, and soil and ocean emissions
(Gettelman et al., 2019). Sulfur emissions for explosive volcanic eruptions are based on version 3.11 of Volcanic Emissions
for Earth System Models (Neely, R. R. and Schmidt, 2016). For Mt Pinatubo, we tested an updated $SO_2$ injection profile over
a larger region and time window than previously used. The new sulfur injection file has been developed because it reproduces
observations better, as described in Section 4.1. In addition, sensitivity simulations using CARMA have been performed to
evaluate differences between the two nucleation schemes used in CARMA and MAM4 and larger sulfur injection amounts that
show improved agreement with observations (Fisher et al., 2019). For the period between 2001 and 2020, we use CAMSv5.1
anthropogenic emissions and biomass burning emissions derived from QFED $CO_2$ fields, multiplied by the species emis-
sions factors collated in Fire INventory from NCAR Version 1.5 (FINNv1.5; Table S1 at http://bai.acom.ucar.edu/Data/fire/;
Wiedinmyer et al., 2011). As described above, DMS ocean emissions, sea salt, and dust emissions are derived internally for
WACCM-MA and CAMchem.



## 4   Stratospheric Aerosol Model Performance

The Mt Pinatubo volcanic eruption in June 1991 was the largest eruption within the last 50 years and is often used to evaluate the performance of ESMs. Stratospheric aerosol optical depth and extinction from satellite observations are available over this
period and are therefore useful measures to test the production and evolution of stratospheric aerosol in the model for a given injection of $SO_2$ after Mt Pinatubo, and also for other smaller eruptions. However, uncertainties exist in the total amount of sulfur injection after volcanic eruptions. Mills et al. (2016) and Mills et al. (2017) have found that the best agreement with optical observations following the Mt Pinatubo eruption using a modal aerosol model occurs with an injection of 10 Tg of $SO_2$. Direct observations (e.g., Fisher et al., 2019) on the other hand suggest that 12-13 Tg of $SO_2$ were present as late as six
days after the eruption. Carn et al. (2016) argue that to find the proper injection amount, one must extrapolate the $SO_2$ back to the initial injection date, yielding as much as 17-19 Tg of $SO_2$ injected. Unfortunately, the chemistry converting $SO_2$ to sulfate and gas-phase reactions that can recycle vapor phase $H_2SO_4$ back to $SO_2$ are uncertain, especially in the first few days due to heterogeneous reactions on ash, which are often not included in models (Zhu et al., 2020). Furthermore, the resolution of ESMs is too coarse to resolve the small-scale plume evolution and the dilution of injected materials in the first day or two, which
influences aerosol microphysical processes. Given the lack of heterogeneous chemistry on ash in these models it is difficult to know how much sulfate aerosol was created by gas phase $SO_2$ chemistry and therefore exactly how much sulfur injections should be used for such a model.

Here, we investigate different model experiments using both aerosol microphysical schemes (MAM4 and CARMA) in the same WACCM-MA setup and different injection amounts, locations, altitude ranges, and aerosol nucleation schemes (see Table
3). Model results are compared to stratospheric aerosol optical depth (SAOD) and aerosol extinction at 525 nm wavelength from the Global Space-based Stratospheric Aerosol Climatology (GloSSAC) for stratosphere aerosol properties (Thomason et al., 2018) (Figures 2, 3, and 6). In addition, we compare SAOD from the Advanced Very High Resolution Radiometer (AVHRR/2) space-borne sensor to the model simulations, which was gridded on a one-by-one degree grid and averaged over different months, as described in Quaglia et al. (2023). Since AVHRR/2 mainly covers the Tropics and has limited coverage between
70°N and 70°S, we are not using the data for comparisons of mid to high latitude averages (Figures 2 and 6). As discussed in earlier work (e.g., English et al., 2012), the GloSSAC dataset underestimates SAOD in the first few months after the eruption compared to AVHRR/2.

### 4.1   Importance of the details of Mt Pinatubo injection locations

Recent model studies using MAM4 (e.g., Mills et al., 2016, 2017; Gettelman et al., 2019) used a single-column $SO_2$ injection
profile to simulate the Mt Pinatubo eruption and injected 5 TgS (equivalent to 10 Tg $SO_2$) between 18 and 20 km at 15°N and 120°E on June 15, 1991. This injection amount was utilized to maximize the agreement between global aerosol optical properties using WACCM6 MAM4 and observations, while recent observational studies suggest larger injection amounts (see above). As expected, when using the same injection profile for WACCM-MA with MAM4 (Figure 2, left top panel, red line), the global distribution of SAOD is within the range of the standard deviation of the GloSSAC and AVHRR/2 observations,





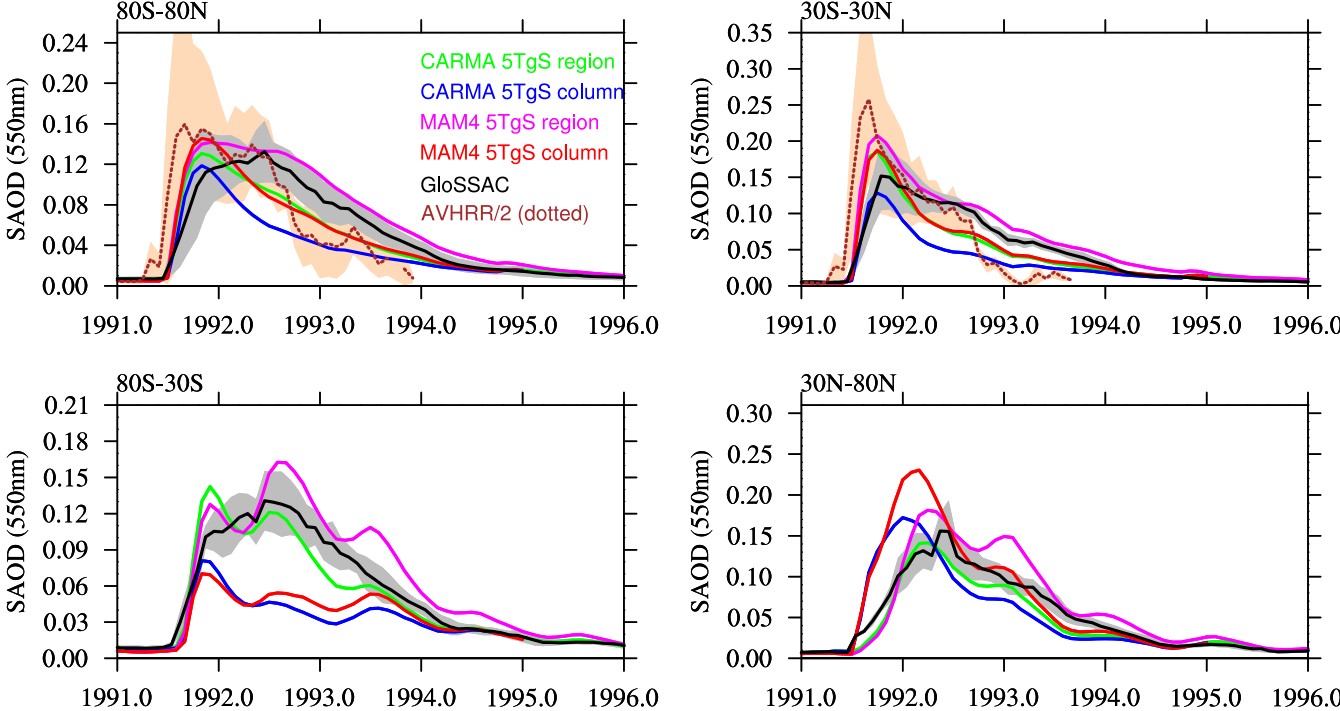

**Figure 2.** Stratospheric Aerosol Optical Depth (SAOD) of different WACCM-MA model experiments using injections in a single column and regional injections using MAM4 and CARMA (see legend) for four different latitudinal averages (different panels) in comparison to GloSSAC and AVHRR/2 data (only available between 70°N and 70°S and therefore only shown for the top panels). Grey and tan areas indicate the two sigma standard deviations of the observational datasets for the corresponding region.

averaged between 80°N and 80°S. However, there is a significant underestimation of SAOD in the Southern Hemisphere (SH) (Figure 2, bottom left, red line), while the Northern Hemisphere (NH) values are within the error bar of the observations. On the other hand, WACCM-MA using CARMA shows a significant underestimation of global SAOD after the Mt Pinatubo eruption using the same single-column injection, and SAOD in both the SH and NH is significantly underestimated (Figure 2, blue line). Based on AVHRR/2 data, both WACCM-MA MAM4 and CARMA underestimate the initial SAOD peak in the Tropics given
a 5 Tg injection of sulfur. Comparisons of WACCM-MA CARMA using a more realistic sulfur injection amount are discussed in Section 4.2.

Aerosol extinction comparisons between GLoSSAC and model simulations with single-column injections averaged between January and March 1992 (Figure 3, middle panels), show that single-column injections in both MAM4 and CARMA result in a spread of aerosols primarily towards the high NH latitudes, while observations also show a spread of aerosols in the SH
lowermost stratosphere 6-9 months after the eruption. Both MAM4 and CARMA show the largest aerosol extinction in the NH polar region lower stratosphere, which points to a too strong transport of aerosols towards the NH high latitudes. Some of the enhancement of aerosol extinction in the SH below 13 km, as shown in observations and models, is the result of the eruption





of Cerro Hudson, which erupted August 15th, 1991, at 45°S and 72°W and injected 2.6 Tg SO$_2$ into the SH mid-latitudes and about 0.75 Tg of SO2 between 12 and 16 km (Carn et al., 2016).

An earlier study by English et al. (2012) showed a much better agreement of aerosol properties after the Mt Pinatubo eruption, using WACCM Version 4 and CARMA. In that study, English et al. (2012) assumed an injection region that covered 2°S-14°N, 95-115°E, between 15-28.5 km over 48 hours (with a peak at 21 km) and used injections of 10 TgS (double the amount used here and towards the high side of observations). They identified the injection region based on observations of the Total Ozone Spectrometer on June 16, 1991. Comparisons of this earlier model study with satellite observations showed a good

representation of SAOD (English et al., 2012). However, this model version did not include the coupling between aerosols and radiation, which may have led to shortcomings in the transport of aerosols after the eruption. Based on these considerations, we developed a new injection profile for the Mt Pinatubo eruption that covers a region between 5°S-15°N, 120°E, and an altitude profile between 19-27 km over 9 hours (with a peak at 22 km) and an initial amount of 5 TgS and 7 TgS (as discussed below). We use injection altitudes above 19 km to ensure that most of the aerosols were directly emitted into the stratosphere,

which allowed for a smaller injection amount than used by English et al. (2012) and expanded the injection region slightly horizontally to allow more aerosol movement into the SH.

    The updated injection details for the Mt Pinatubo eruption result in an improved agreement of extinction and SAOD with GloSSAC using both MAM4 and CARMA configurations (Figures 2 and 3). In particular, the updated injection region and timing improved the transport of aerosols toward the SH (Figure 3, right columns). While WACCM-MA MAM4 captures the

peak and decline of SAOD very well in the Tropics, it shows a slight overestimation in both NH and SH compared to GLoSSAC (Figure 2, magenta lines). WACCM-MA CARMA shows a substantial improvement in SAOD compared to a single-column injection for both Tropics and mid-latitudes (Figure 2, green lines). SAOD values in the Tropics are within the standard deviation of the AVHRR/2 observations and GLoSSAC observations for the peak value, and the model agrees within the error bars of the observations in the NH and SH. Additional improvements of WACCM-MA CARMA compared to observations,

including changes to the nucleation scheme and injection amount are discussed in Section 4.1.2.

    To analyze the differences between MAM4 and CARMA, we compare the evolution of SO$_2$ and sulfate aerosol burden and other relevant variables in the volcanic plume within the first 30 and 210 days after the Mt Pinatubo eruption (Figure 4). The volcanic plume is defined here as locations within the stratosphere (60°S-60°N and 10-150 hPa) and for grid points that exceed 0.1 um$^2$/cm$^3$ surface area density (SAD). We limit the region of interest to airmasses within the volcanic plume and

exclude the polar region and tropospheric air masses. The main difference between the injection in a single-column and the larger region (as described above) is a much larger limitation of OH in the first month for both MAM4 and CARMA for the single-column injection. This is likely because the SO$_2$ is diluted in the regional case and not able to reduce OH in the same way as the single-column injection case (Figure 4, top right panel).

    H$_2$SO$_4$ is formed through the oxidation of SO$_2$ and is therefore dependent on the available OH that is somewhat smaller for

CARMA than for MAM4. The nucleation of sulfate from sulfuric acid gas forms small initial sulfuric acid particles (or sulfates) that build up in the smallest pure sulfate bins for CARMA, while the coagulation of similar-sized particles is suppressed due to a low coagulation kernel. In MAM4, the Aitken mode, which is much larger than the smallest bin in CARMA, serves as a large





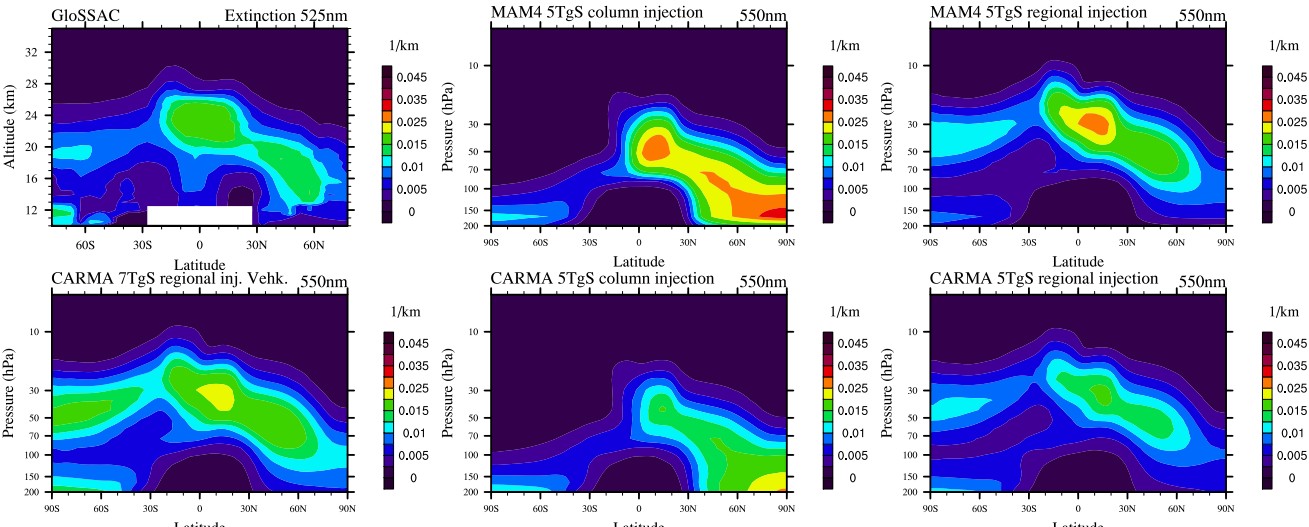

**Figure 3.** Zonal average aerosol extinction (550 nm for the model and 525 nm for GLoSSAC), averaged between Jan - March 1992 for the GlosSAC climatology (top left) and different model simulations using WACCM-MA.

particle target producing a much larger coagulation kernel with the molecules. Therefore, the acid molecules are rapidly lost from the gas phase and nucleate much faster. The initial larger coagulation and growth produce more sulfate in MAM4 than in

CARMA. In contrast in CARMA, $H_2SO_4$ builds up in the first two days of the eruption and then slowly declines, while sulfate aerosols are nucleating and also condensing on existing particles within the first 30 days in the volcanic plume. Consequently, the effective radius is initially smaller using CARMA compared to MAM4. The initial very small effective radius in CARMA, most pronounced for the 1-column injection, leads to an initial large peak in SAD in the first day or two and a later decline below the SAD in MAM4 after about five days. For both aerosol models, injections in one column result in a smaller effective

radius and sulfate mass than regional injections in the first two months due to the initial OH limitation for the one-column injections (Figure 4, middle panel).

Differences between single-column and regional injections and between MAM4 and CARMA are also reflected in the $SO_2$ lifetime (Figure 5). The single-column injection results in an e-folding time of $SO_2$ of 39 days for MAM4 and 52 days for CARMA, while the regional injection cases show reduced lifetimes of 36 and 45 days, respectively. The longer $SO_2$

oxidation lifetimes delay the production of sulfuric acid gas ($H_2SO_4$) in the single-column injection (Figure 4, top middle panel). Differences in the lifetimes between MAM4 and CARMA are likely a result of differences in the recycling of $SO_2$ from sulfuric acid ($H_2SO_4$) through photolysis of $H_2SO_4$ and $SO_3$, which strongly increases in CARMA in the first day after the volcanic eruption (Figure 4, top middle panel).

A few weeks after the eruption, the effective radius in CARMA grows larger than in MAM4 and reaches between 0.4 to 0.5

microns after three months of the eruptions in CARMA, which is in very good agreement with SAGEII observations, as shown in English et al. (2012). MAM4 effective radius stays below 0.4 microns. A better representation of aerosol size in CARMA is



**Figure 4.** Timeseries of airmasses in the volcanic plume between 60°S and 60°N and between 10 and 150 hPa, defined by grid points with stratospheric surface area density larger than 0.1 $\mu$m2/cm$^3$ using daily averaged model output for different chemistry and aerosol variables, over the first 30 days (first and second row) and over the first six and half months (bottom row), comparing WACCM-MA experiments with injections in a single grid box (column) and regional injections using CARMA and MAM4.

expected due to a more comprehensive microphysical scheme using a sectional aerosol model. Furthermore, SAD in MAM4 is consistently larger than in CARMA, corresponding to the smaller effective radius for a similar or larger mass. The total sulfate mass in CARMA declines slightly faster than for MAM4, which is likely the result of stronger sedimentation of larger

particles and removal outside the considered region (between 60°N and 60°S). This is particularly true for the single-injection case, where sulfate aerosols move faster towards the NH high latitudes than the regional injection case (as suggested in Figure 4, middle panels).

## 4.2  Comparisons of different nucleation schemes and Mt Pinatubo injection amount in CARMA

Comparisons in 4.1. have been performed with the standard nucleation schemes for MAM4 (the Vehkamäki scheme) and

CARMA (the Zhao scheme), see Section 2.3 for more details. Here we are exploring possible differences between MAM4 and CARMA that may be caused by differences in using the nucleation scheme (Figure 6 and Figure A1). Using the regional



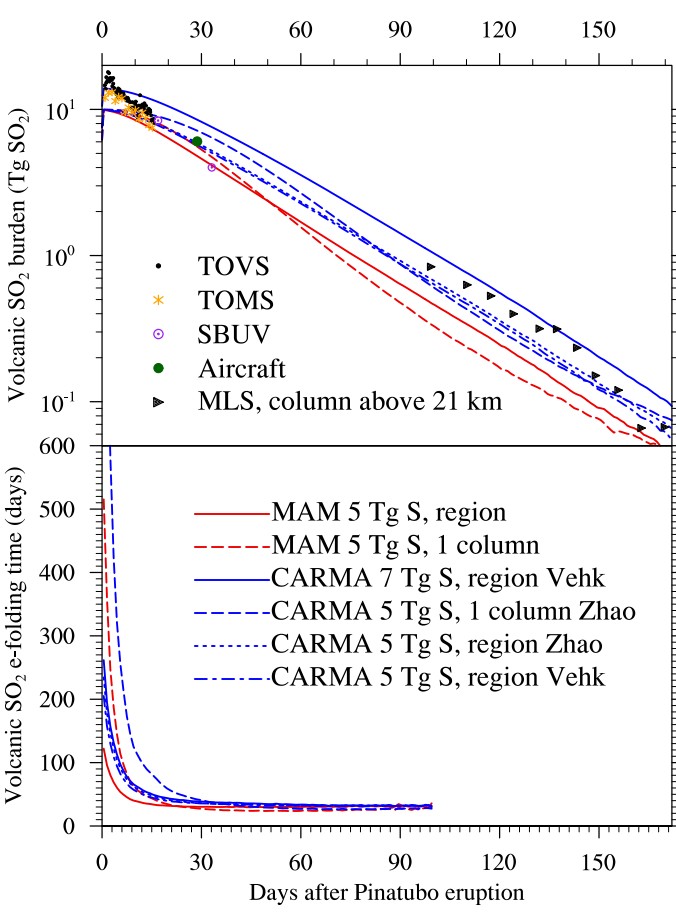

**Figure 5.** Top panel: Calculated global volcanic $SO_2$ burden following the 15 June 1991 eruption of Mount Pinatubo is compared to observations. The lines show the daily average global burden of $SO_2$ calculated for the WACCM-MA CARMA (blue) and the WACCM-MA MAM4 (red) simulations minus the $SO_2$ burden calculated in corresponding simulations that exclude the Mt Pinatubo eruption. Observations from TOVS (black circles) and TOMS (orange asterisks) show an initial burden of 13–18 Tg $SO_2$, of which 10 Tg remained after loss to sedimentation ice and ash in the first 7–9 days (Guo et al., 2004). Observations from SBUV, aircraft and MLS are shown as presented in Read et al. (1993). Bottom panel: Volcanic $SO_2$ e-folding time (days) shown as a function of days following the 15 June 1991 eruption of Mt Pinatubo in the simulations. The e-folding time is derived from the daily change in the global volcanic $SO_2$ burden. Volcanic $SO_2$ is calculated by subtracting the global burdens from corresponding simulations that exclude the Mt Pinatubo eruption.



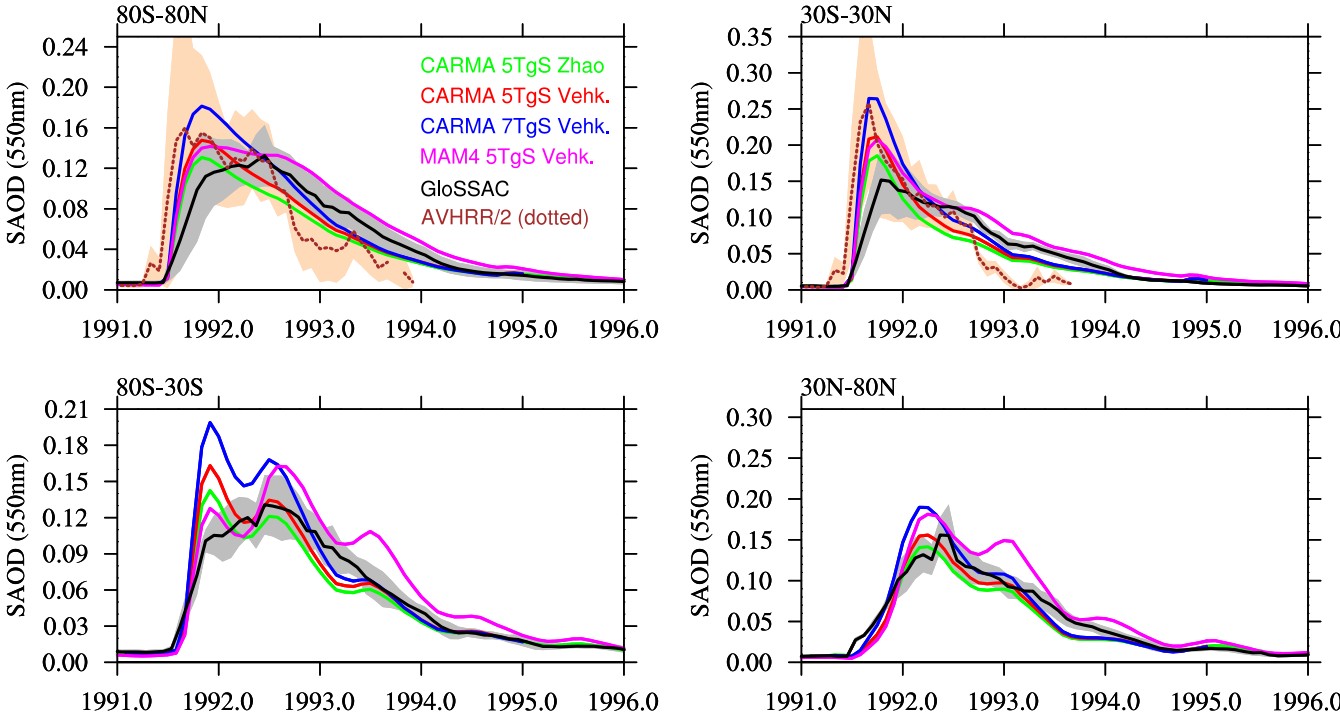

**Figure 6.** Stratospheric Aerosol Optical Depth (SAOD) of different WACCM-MA model experiments with regional injections using different nucleation schemes and injection amounts for MAM4 and CARMA (see legend), illustrated for four different latitudinal averages (different panels) in comparison to GloSSAC and AVHRR/2 data (only available between 70°N and 70°S and therefore only shown for the top panels). The default nucleation scheme for CARMA is the Zhao scheme (green colors), and for MAM4 it is the Vehkamäki scheme (magenta). Using the Vehkamäki scheme in CARMA for different injection amounts is shown in red (5TgS) and blue (7TgS). Grey and tan areas indicate the two sigma standard deviations of the observational datasets for the corresponding region

injection profile and injections of 5 TgS, we performed two model simulations using WACCM-MA CARMA, one using the original nucleation Zhao scheme and a second with the Vehkamäki scheme, consistent with what has been used in MAM4. In addition, we also tested simulations that increase the injection amount to 7 TgS using CARMA, which is more in line with an
updated observational study by Fisher et al. (2019), who suggested even larger sulfur injections of up to 8 or 9 TgS. However, some initial sulfur injection amount is expected to be removed early by reactions with volcanic ash (Zhu et al., 2020), which is not included in this model version.

Using the Vehkamäki scheme with CARMA shows a slight increase in SAOD compared to using the Zhao scheme (Figure 6, green and red lines). The additional larger injection of 7 TgS compared to the 5 TgS (Figure 6, blue lines) results in a larger
SAOD peak in the Tropics, which is more in line with what has been observed by AVHRR/2. It is close to or within the range of the GloSSAC standard deviation during the decline of the plume for the different regions. Comparisons of different relevant species in the volcanic plume in the first 30 and 210 days after the eruption (Figure A1) indicate that the Vehkamäki scheme





results initially in slightly larger $H_2SO_4$ and does not show the initial peak in SAD, which points to slightly larger nucleation and faster condensation and slightly reduced recycling of $SO_2$ than the Zhao scheme. Using the Vehkamäki scheme compared
to the Zhao scheme does not significantly impact the $SO_2$ lifetime, which is reduced from 45 to 44 days. The more considerable injection amount using 7 TgS instead of 5 TgS increases available sulfur and $H_2SO_4$ for condensation, resulting in a similar sulfate burden compared to MAM4 for the first month after the eruption and a much larger burden of the peak after 2-3 months.

### 4.2.1   Particle Number density distribution comparisons between CARMA and MAM4

Based on the above analysis, WACCM-MA with both MAM4 and CARMA are able to reproduce the observed SAOD evolution
after the Mt Pinatubo eruption if specific injection regions and amounts are applied. MAM4 needs a 5 TgS injection to agree well with the observations in the Tropics, with some overestimation of AOD in the mid to high latitudes. CARMA does better with larger injections of $SO_2$ in the range of 7 TgS, which agrees with $SO_2$ observations. However, the effective radius and surface area density are different between the different configurations. In order to evaluate differences in the simulated size distribution between the two models, we compare the model result to the accumulated particle number density distribution from
the Laramie Wyoming balloon particle counter (Deshler et al., 2003, 2019) at 20 km altitude for two different periods after the Mt Pinatubo eruption in October 1991 and March 1992 (Figure 7). The accumulated particle number density distribution is a direct measurement. Each column or symbol represents the number density of particles larger than certain sizes. CARMA and MAM4 simulations use the Vehkamäki nucleation scheme and 5 TgS injections for the Mt Pinatubo eruptions (the 7 TgS injection case for CARMA is shown in Figure A2). For CARMA, pure sulfate, and mixed aerosol groups are shown as blue
and red histogram plots. We also derive the number size distribution and volume size distribution per radius for both CARMA and MAM4 to identify differences in where the aerosol mass is distributed.

The CARMA particle number density reproduces the accumulated number distribution compared to observations quite well after three and nine months following the Mt Pinatubo eruption, besides an overestimation of the accumulated number density for the largest bin around 1 $\mu$m for both considered periods. The accumulation of particles in the largest CARMA bin indicates
that the selected range for pure sulfates may not be sufficient to reproduce observed aerosol distributions. On the other hand, MAM4 also overestimates the number densities for 1 $\mu$m, and number densities are underestimated between 0.1 and 0.4 $\mu$m, and overestimated for 0.01 $\mu$m compared to the balloon observations. MAM4, therefore, overestimates the total number of smaller particles than observed after Mt Pinatubo, while CARMA shows a better agreement with observations. Number and volume size distributions (Figure 7, middle and bottom panel) support that MAM4 underestimates the number of accumulation
mode particles and overestimates Aitken mode particle number compared to CARMA. Furthermore, the narrow coarse mode peak in MAM4 cannot reproduce observed aerosol sizes near 0.4 $\mu$m, where most of the mass is located. This is aligned with a smaller effective radius in MAM4 compared to CARMA and observations. The smaller particles and larger sulfate burden in MAM4 are also aligned with a larger surface area density (described above) and larger SAOD in MAM4 than CARMA (consistent with Figure 4), which can have implications for stratospheric chemistry (not investigated here). Experiments using
injections of 7 TgS for simulations with CARMA using the Vehkamäki and the Zhao (not shown) nucleation schemes show very similar size distributions (Figure A2 for Vehkamäki).



**Figure 7.** Accumulated particle number density size distribution (top), number size distribution (dN/dlogr) (middle), and volume size distribution (dV/dlogr) (bottom) comparisons of different model experiments compared to Wyoming balloon observations (black circles) at 20 km for October 1991 (left) and March 1992 (right) after the eruption of Mt Pinatubo. Error bars of the observations are, for the most part, smaller than the illustrated symbol. Monthly averaged model results for different experiments using CARMA and MAM4 with regional injections of 5 TgS (see more details in the text) and the Vehkamäki nucleation scheme. CARMA size distributions for the mixed group are shown in red and for the pure sulfate group in blue. MAM4 particle size modes are shown as green lines.





### 4.3 Stratospheric optical and aerosol properties and total column ozone between 2001-2020

After the large Mt Pinatubo eruption in 1991, we experienced a volcanically quiet period until early 2000. Surface area density and particle number density distribution for the volcanically quiet period of the different experiments are compared to the

balloon particle counter in July 2003 at Laramie, Wyoming, at 20 km altitude (Figure 9). CAMchem and WACCM-MA using CARMA with the Zhao and the Vehkamäki nucleation scheme reproduce SAD from observations very well. CAMchem and WACCM-MA using MAM4 somewhat overestimate the mean SAD compared to the observations in particular CAMchem, which is mostly outside the error range of the observation. The uncertainty of SAD from the observations is about 40% (Deshler et al., 2003) because SAD is a derived measure from the observed size distributions. Comparing the accumulated

particle number density distribution allows a more direct comparison between observations and model results (as performed in Figure 9, middle and bottom row). CAMchem and WACCM-MA using CARMA agree with the observations for most size bins, with a slight overestimation in the number for the two largest bins, which is more pronounced using WACCM-MA. For CAMchem, the Zhao nucleation scheme shows a slightly better agreement with observations, which is not the case for WACCM-MA. In contrast, configurations using MAM4 show an overestimation of the aerosol number for sizes larger 0.4 $\mu$m,

which is part of the MAM4 coarse mode. Observations also indicate an overestimation in the number densities of the smallest aerosol size, the Aitken mode in MAM4, which is more pronounced in CAMchem. The larger number of smaller aerosol particles is likely responsible for the larger SAD shown in Figure 9 (left panel). In the following, we only discuss results using the Zhao nucleation scheme for CARMA. As shown later, CARMA with Zhao performs better than using the Vehkamäki nucleation scheme in the troposphere compared to observations.

After 2000, a series of smaller volcanic eruptions emitted up to 1 TgS each, increasing SAOD (Santer et al., 2014). WACCM-MA and CAMchem configurations reproduce the global annual mean SAOD evolution within the standard deviation of GLoSSAC (Figure 8). For background conditions, model experiments using MAM4 show a slight underestimation compared to GLoSSAC climatological mean, while simulations with CARMA are very close to the observed values. All the experiments show a relative overestimation of the peak values, particularly for the Kasatochi eruption in 2008 and other larger eruptions.

Reasons for the overestimation of these volcanic eruptions may be a result of the specifics of the volcanic emission database Neely, R. R. and Schmidt (2016), or it may be due to $SO_2$ interactions with ash and ice that are mixing in the simulations, which will have to be investigated in future studies.

In addition to the evaluation of aerosol properties, we also performed a comparison of total column ozone (TCO) in both the stratosphere (Figure 10) and troposphere (Figure A3) with the MLS/OMI climatology between 2004 and 2010. All model

configurations generally reproduce the zonal structure and seasonality of observed stratospheric TCO, with fairly good agreement in the Tropics and mid-latitudes. All the models show an underestimation of stratospheric TCO in the SH mid-and high latitudes and some underestimation in NH mid-to-high latitudes in July and October. This behavior has also been identified by Davis et al. (2023), and is likely a result of insufficient ozone transport from the Tropics to the high latitudes. CAMchem shows slightly larger TCO values in January and April in the NH high latitude, indicating stronger transport to the NH for CAMchem.

No significant differences can be identified between MAM4 and CARMA.



**Figure 8.** Surface Area Density (top) and accumulated particle number density size distribution comparisons of monthly averaged model results for different experiments (middle and bottom) with CAMchem (left) and WACCM-MA (right) compared to Wyoming balloon observations for stratospheric aerosol background conditions at 20 km in July 2003. Top panel: For CARMA, results based on the Zhao (solid lines) and Vehkamäki (dashed lines) nucleation scheme are shown. Middle panel: Experiments used the Zhao nucleation scheme for CARMA and the Vehkamäki nucleation scheme for MAM4. Bottom panel: All experiments used the Vehkamäki nucleation scheme.

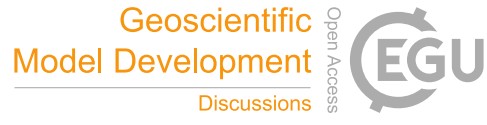

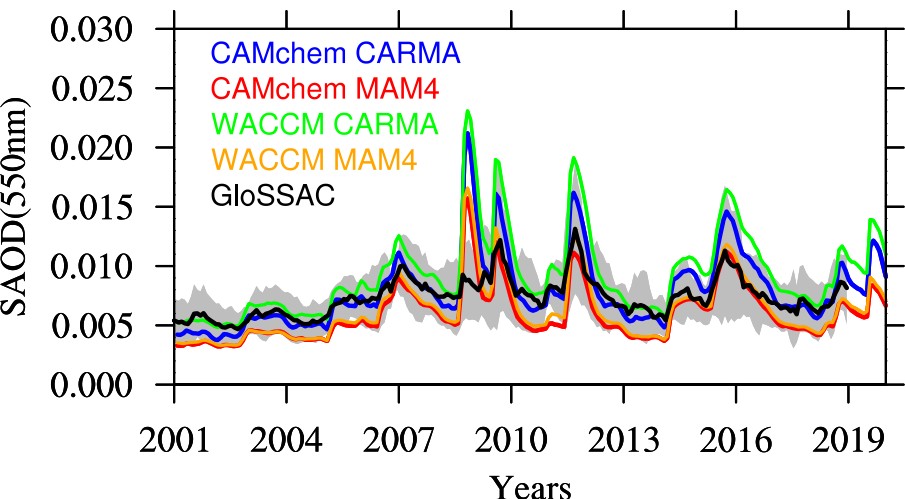

**Figure 9.** Global mean stratospheric AOD (SAOD) comparisons between different model experiments (see legend) between 2001 and 2020, compared to GLoSSAC. The grey area indicates the two sigma standard deviations of the observational dataset. CARMA model experiments use the Zhao nucleation scheme, and MAM4 experiments use the Vehkamäki nucleation scheme

.

## 5 Tropospheric Aerosol Model Performance

CAMchem includes interactive aerosols in both the troposphere and stratosphere and, as a low-top model, is more suited for studies focusing on the UTLS and the troposphere. CAMchem simulates oxidants and ozone well (Emmons et al., 2020) and shows a reasonable agreement with tropospheric TCO compared to OMI observations (Figure A3).

The following evaluations will focus on CAMchem. However, aerosol burdens and budgets will also be compared with WACCM-MA. We only evaluate the general performance of the model using climatological and background aerosol quantities for the troposphere. A more detailed evaluation of specific case studies will be done in future studies. For this, we focus on two data sets. The first is AOD in the visible (550nm) from satellite observations from the Moderate Resolution Imaging Spectroradiometer (MODIS) sensors on both the Terra and Aqua platforms, based on the combined Dark Target/Deep Blue

AOD algorithm, version 6.1, documented in Levy et al. (2013). We derived a climatology between 2001 and 2019 for different seasons. We also compare to a climatology derived from MERRA2, a reanalysis product that includes the assimilation of trace gases and aerosols (Randles et al., 2017; Buchard et al., 2017). The second dataset we use is from the NASA Atmospheric Tomography Mission (ATom) aircraft mission (Wofsy, 2018). This dataset is currently the most comprehensive aircraft dataset available, including chemical and aerosol properties information. Flights sampled vertical profiles in each of the four seasons

(ATom1-4) over a 3-year period between 2016 and 2018. The dataset covers an area from California northward to the western Arctic, then southward to the South Pacific, eastward to the Atlantic, northward to Greenland, and then returns to California across central North America. Based on this dataset, a comprehensive aerosol properties dataset has been derived (Brock et al., 2021). This aerosol properties dataset includes information on aerosol microphysical, chemical, and optical properties derived





**Table 4.** Averaged total (and tropospheric for sulfate) aerosol burden for 2001-2002 background conditions. All numbers are provided in Tg for burdens and Tg/yr for the emissions, dry and wet deposition, chemical and aqueous-phase productions, and net gas-to-aerosol exchange. The numbers for sulfate are given in TgS for the burden and TgS/yr for the other quantities. The lifetime is given in years.

| Model | | CAMchem | WACCM-MA | CAMchem | WACCM-MA |
| Aerosol | | CARMA | CARMA | MAM4 | MAM4 |
|---|---|---|---|---|---|
| Sea Salt | Burden | 3.1 | 3.5 | 7.0 | 7.2 |
| | Emissions | 7302 | 6701 | 3139 | 3116 |
| | Dry Deposition | 5190 | 4161 | 657 | 657 |
| | Wet Deposition | 2106 | 2537 | 2382 | 2340 |
| | Lifetime | 0.2 | 0.2 | 0.9 | 0.9 |
| Dust | Burden | 12.3 | 12.4 | 31.3 | 22.3 |
| | Emissions | 9125 | 6116 | 3048 | 2026 |
| | Dry Deposition | 8596 | 5569 | 990 | 703 |
| | Wet Deposition | 515 | 544 | 2039 | 1313 |
| | Lifetime | 0.5 | 0.7 | 3.8 | 4.0 |
| Black Carbon | Burden | 0.09 | 0.08 | 0.12 | 0.13 |
| | Emissions | 8.5 | 8.5 | 8.5 | 8.5 |
| | Dry Deposition | 3 | 3 | 2 | 3 |
| | Wet Deposition | 5 | 6 | 6 | 6 |
| | Lifetime | 3.9 | 3.7 | 5.3 | 5.6 |
| Primary Organics | Burden | 0.48 | 0.45 | 0.64 | 0.71 |
| | Emissions | 45 | 45 | 45 | 45 |
| | Dry Deposition | 17 | 16 | 12 | 14 |
| | Wet Deposition | 28 | 29 | 32 | 31 |
| | Lifetime | 3.9 | 3.7 | 5.3 | 5.8 |
| Secondary Organics | Burden | 1.15 | 1.15 | 1.07 | 0.78 |
| | Net Gas-Aerosol | 131 | 200 | 138 | 77 |
| | Photolysis | 68 | 57 | 63 | 0 |
| | Dry Deposition | 8 | 30 | 8 | 10 |
| | Wet Deposition | 55 | 113 | 67 | 65 |
| | Lifetime | 6.8 | 2.9 | 5.2 | 3.8 |
| Sulfate (total) | Burden | 0.71 | 0.64 | 0.60 | 0.56 |
| Sulfate (trop.) | Burden | 0.56 | 0.48 | 0.41 | 0.40 |
| | Aq-phase Chemistry | 15 | 10 | 17 | 13 |
| | Chemical Production | 15 | 17 | 14 | 16 |
| | Vertical Emissions | | | 0.3 | 0.3 |
| | Dry Deposition | 5 | 4 | 4 | 3 |
| | Wet Deposition | 25 | 23 | 28 | 26 |
| | Lifetime (trop.) | 6.8 | 6.5 | 4.7 | 5.0 |



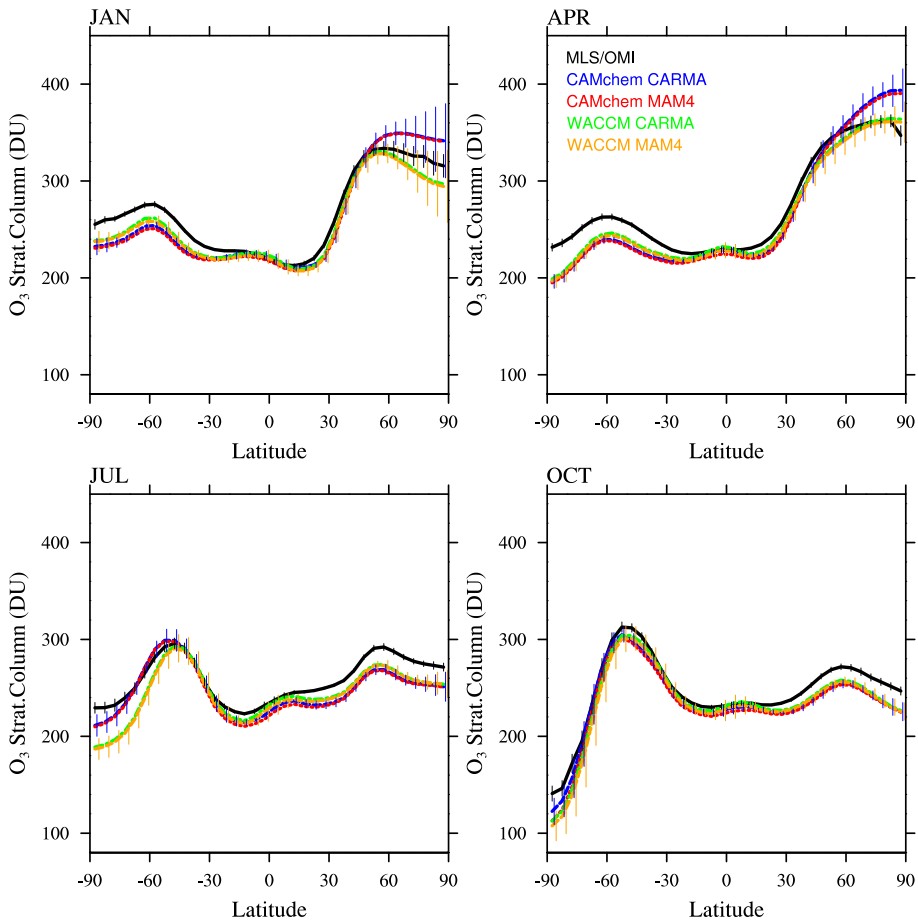

**Figure 10.** Monthly and zonally averaged stratospheric ozone column (in DU) comparison between OMI/MLS observations between 2004 and 2010 (black) and different model experiments between 2004 and 2010 (for ozone <150 ppb in the model), for four months. OMI/MLS error bars (black) show the zonally averaged $2\sigma$ 6-year root-mean-square standard error of the mean at a given grid point, derived from the gridded product (Ziemke et al., 2011). Model results are interpolated to the same $5°$ latitude grid as the observations, the error bar the standard deviation ($1\sigma$) of the interannual variability per latitude interval for CAMchem CARMA and WACCM-MA MAM4.

for both dry and ambient conditions from in-situ measurements made during the four ATom campaigns. Here, we use the
composition-resolved size distributions that range from 3 nm to 50 $\mu$m in diameter.

### 5.1 Tropospheric Aerosol Optical Depth

Both CAMchem CARMA and MAM4 underestimate total AOD compared to MODIS and MERRA2 in June-July-August (Figure 11). CARMA underestimates AOD by up to 60 % over the ocean and land and overestimates some land regions over South America and Australia. MAM4 underestimates AOD in the NH up to 80% and overestimates the SH by more than 100%
in some regions. Significant overestimation of AOD in MAM4 is shown over land regions in South America, South Africa, Australia, and parts of North East Africa and the Middle East. MERRA2 also overestimates some land regions, including South America and Australia, but shows a much better agreement with observations over the ocean. In December-January-February





(Figure 12), CARMA underestimates AOD over the ocean with more negative values in the Southern part of the Southern Ocean. As for June-July-August, some land regions are overestimated, but the values are comparable to MERRA2, besides

some overestimation over South America. MAM4 over and underestimates AOD for different parts of the ocean and does not show the North-South gradient in AOD differences in December-January-February. The overestimate of AOD over South America, Africa, and Australia remains for December-January-February.

In summary, CARMA shows a stronger underestimation of global AOD compared to satellite observations and MERRA2 as a result of a general underestimation of AOD over the oceans due to missing sources of marine organic aerosols (Yu et al., 2015;

Zhao et al., 2021) as well as missing nitrates in the model (Jo et al., 2021; Lu et al., 2021). The significant overestimation of AOD in the SH in MAM4 is likely a result of too many sea-salt emissions. The significant overestimation of AOD over South America for both aerosols models may be a result of a too strong formation of secondary organic aerosols from organic precursor emissions. The improved representation of AOD in CARMA over Africa and also differences over the ocean are likely a result of different sea-salt and dust emissions parameterizations in the two configurations (see Section 2.4.5.). For

example, in MAM4, the overestimation of AOD in the SH in both JJA and DJF, e.g., Australia, South Africa, and Argentina, is due to an overestimation in the dust emissions (Li et al., 2022).

### 5.2 Comparisons to ATom observations and aerosol budgets

The following section focuses on evaluating the model performance for background aerosol conditions in the troposphere based on ATom 1-4 observations. We use the integrated mass of sulfate, organics, nitrate, sea salt, dust, and black carbon in coarse

and fine fractions and extinction provided by the data set and fit lognormal functions to compare to different aerosol modes for MAM4 and selected size ranges for CARMA, based on 1-minute flight data. We select available data over remote regions over the Pacific (200°E to 145°E) and Atlantic (0° to 80° East) and average over different regions between 60°N and 60°S. The model interpolated the output to the closest location of the flight track of each 1-minute measurement and then averaged over the same region as the observations. Differences between the model configurations are also discussed based on aerosol

budgets, including emissions, deposition, chemical production, and lifetime (Table 4).

#### 5.2.1 Sea-salt and Dust

Differences in sea-salt and dust burdens between CARMA and MAM4 are the result of differences in how emissions are calculated, microphysical parameterization that result in different aerosol size distributions, and resulting differences in the removal. In particular, sea-salt and dust burdens derived from the CARMA are only about half the amount derived using

MAM4, which is a result of differences in tuning choices (Table 4). On the other hand, dry deposition of sea salt and dust is more than five times as large using CARMA, despite the smaller burden. Differences in sea-salt and dust dry deposition are the result of the narrow standard deviation of 1.2 in CESM2 MAM4, which was chosen to accommodate the stratospheric coarse model sulfate (Mills et al., 2016). As discussed in Li et al. (2022), the narrow mode reduces the number of larger aerosol sizes that would experience more dry deposition and, with that substantially reduces the dry deposition of MAM4. The resulting

longer lifetime of both dust and sea salt requires smaller emissions to achieve a similar burden (Liu et al., 2012). The larger



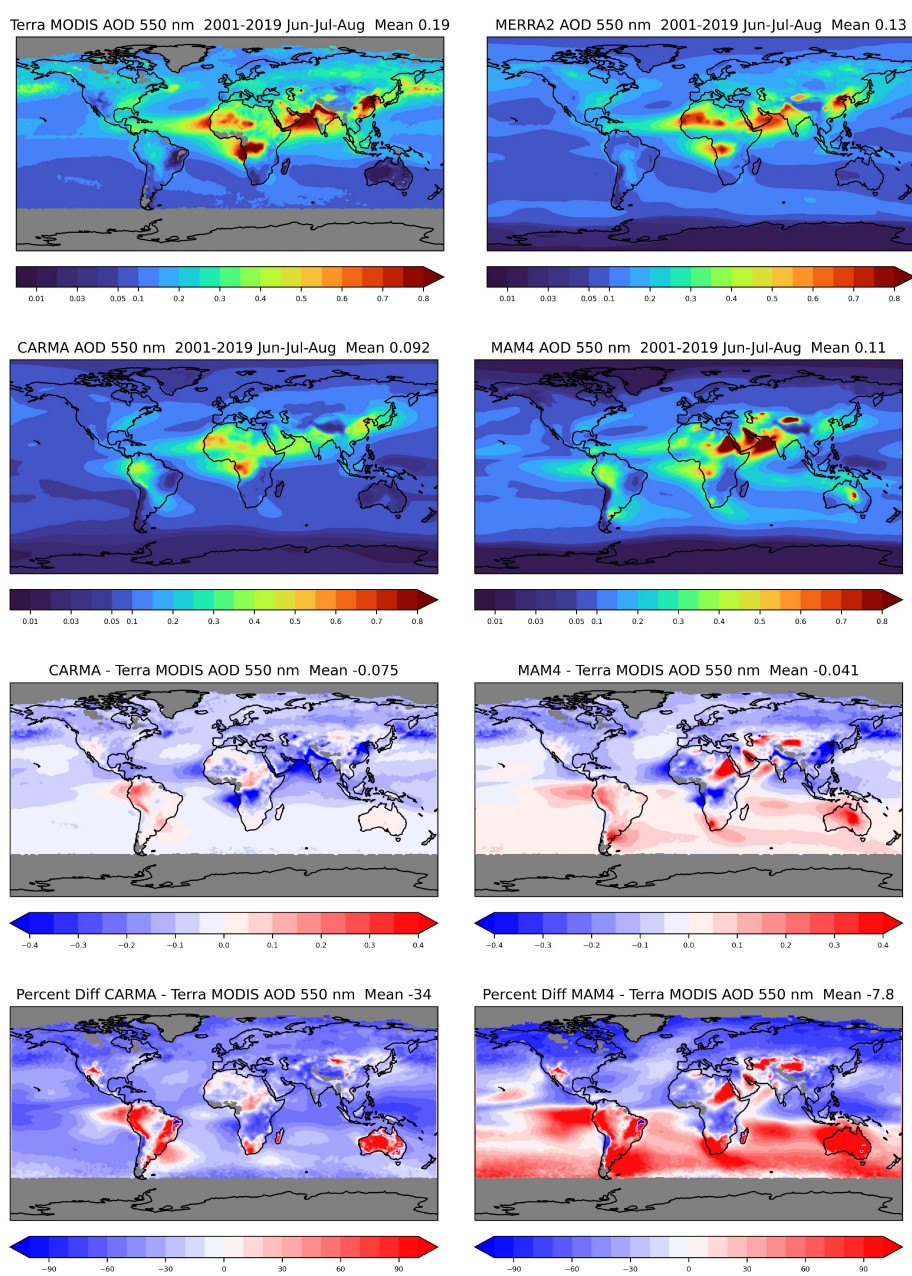

**Figure 11.** Aerosol Optical Depth in the visible (550nm), June, July, August (JJA) averaged between 2001 and 2020 from MODIS (TERRA) observations (top left), MERRA (top right), and from CAMchem CARMA (middle left) and CAMchem MAM4 (middle right). The bottom panels show differences between CAMchem CARMA and the MODIS observations (left) and between CAMchem (MAM4) and observations (right).



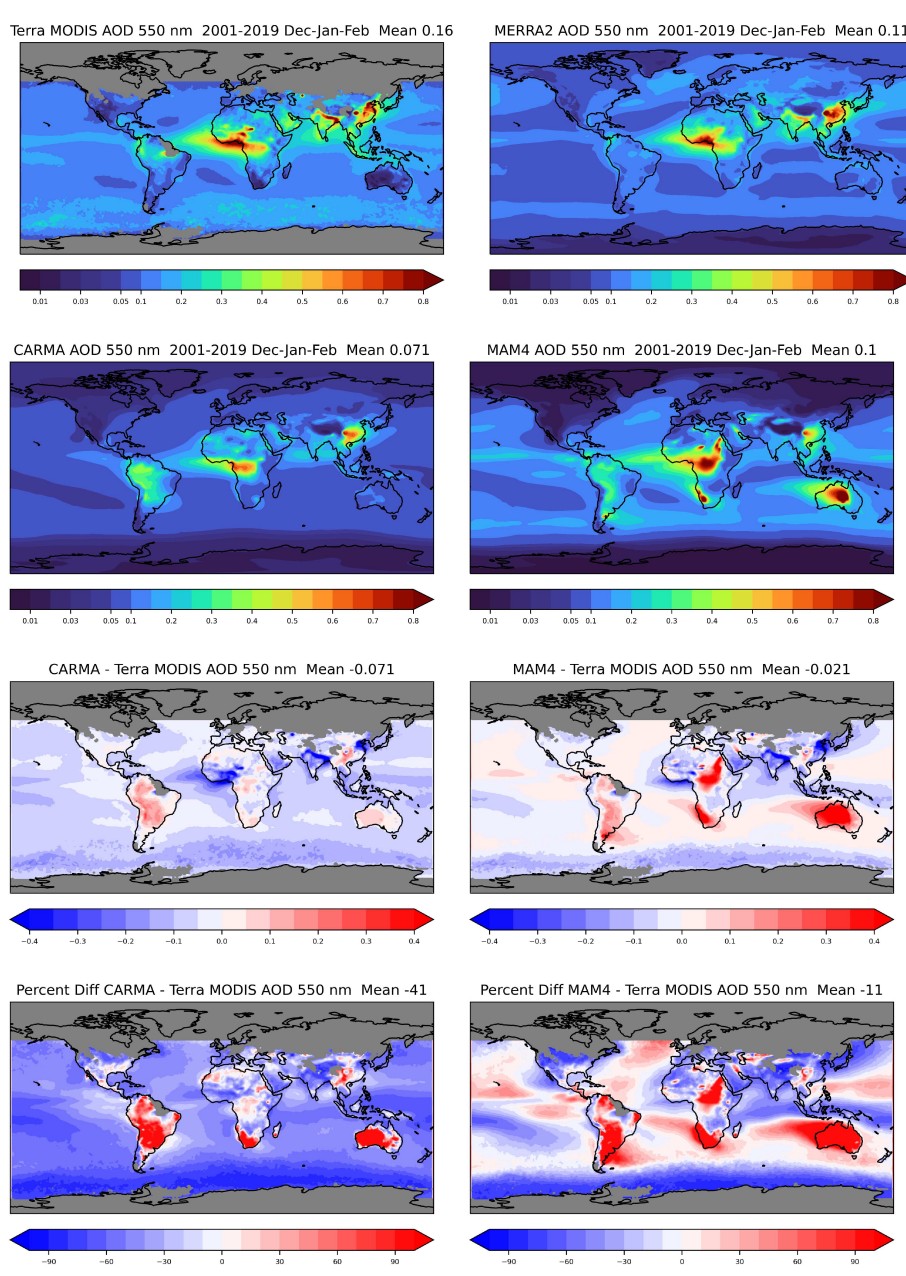

**Figure 12.** As Figure 11, but instead for December, January, February (DJF).

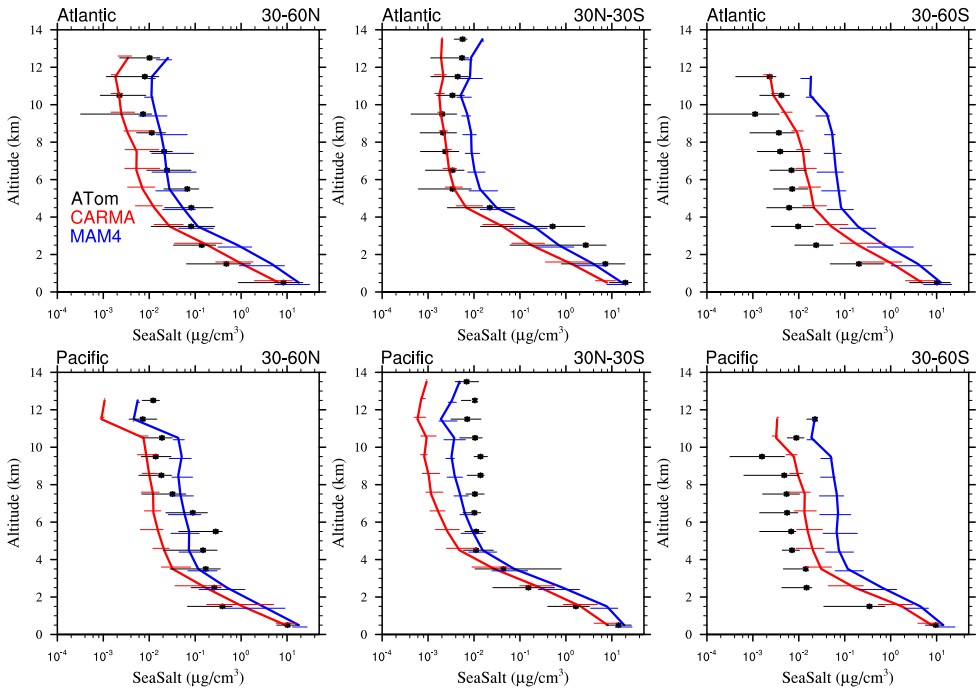

**Figure 13.** Vertical profile comparisons of the median of CAMchem CARMA (red) and CAMchem MAM4 (blue) and ATom1 to ATom 4 aircraft observations, averaged over three regions, 30-60°N, 30°N-30°S, and 30-60°S. The model results were saved on the flight track using the closest location of each 1-minute data point and then averaged over the same regions as the observations. Error bars for the observations indicate the 25th and 75th percentile of the distribution for different regions.

amount of dry deposition in CARMA than MAM is a result of a faster settlement of emissions in larger size bins in CARMA, as discussed in more detail in Yu et al. (2015), who showed that with a similar sea salt burden, both emissions and dry depositions are larger in CARMA compared to MAM4. Wet deposition is fairly similar for sea salt between the two different aerosol schemes. However, since the burden for sea salt in CARMA is about half compared to MAM4, wet removal in CARMA is 580  more efficient. On the other hand, wet removal in CARMA is 2-4 times smaller for dust, which is aligned with the relatively smaller burden in the dust.

Comparisons to sea salt (Figure 11) and dust (Figure 12) with ATom aircraft observations for different regions indicate that both models are within the error bars of the data near the surface for sea salt. CARMA is also within the error bars close to the surface for dust, but MAM4 tends to overestimate dust mass near the surface. The larger AOD over the SH in MAM4 shown 585  in Figures 11 and 12 is partly a result of the overestimation of sea salt and dust in MAM4 in the mid-latitudes of the SH. On the other hand, CARMA underestimates sea salt in the high latitude NH between 4 and 8 km (Figure 13). Furthermore, while CARMA shows a very good agreement with the sea salt observations over the tropical Atlantic, it underestimates sea salt in the tropical Pacific above about 4 km. MAM4 overestimates sea salt above 6 km in the tropical Atlantic and for the SH in both the Pacific and Atlantic. Sea salt has strong vertical gradients. It is likely that convection near the measurement sites influences its



**Figure 14.** As Figure 13, but instead for dust.

abundance, and convection is difficult to reproduce in global models both due to its small spatial scale and its episodic nature. More detailed investigations in the future are needed to identify the regional differences that may be caused by differences in clouds and rainfall. CARMA and MAM4 show a constant offset, with CARMA being somewhat lower than MAM4, likely caused by differences in sea-salt emissions and the stronger dry deposition in CARMA compared to MAM4.

Both CARMA and MAM4 overestimate dust in the tropical Atlantic region by an order of magnitude in the mid-troposphere, while they agree well with observations in the tropical Pacific. This is in contrast to the comparison by Lian et al. (2022), who reported an overestimation over the Pacific and good agreement over the Atlantic, using CESM1(CARMA) and ATom1 for comparisons. CESM2 using CARMA also reproduces dust concentrations in the Southern Hemisphere mid-latitudes well over the Atlantic and Pacific oceans. CARMA and MAM4 further agree very well with the observation for the NH mid-latitudes. The main difference between CARMA and MAM4 is an overestimation of dust in MAM4 by one order of magnitude in the SH mid-latitudes, while CARMA agrees well with the observations. Differences compared to the observations in the tropical Atlantic mid-troposphere may be related to how CARMA and MAM4 apply wet removal. Differences in the SH mid-latitudes





between CARMA and MAM4 are the result of the overestimation of smaller particle emissions and underestimation of dry removal in MAM4, for example, over Australia and South Africa (Li et al., 2022).

### 5.2.2 Black Carbon and Organic Aerosols

605 Black Carbon (BC) and Organic Aerosol (OA) are strongly impacted by both dry and wet removal. With freshly emitted BC being mostly hydrophobic, aged BC experiences wet removal as part of the mixed aerosol particle in CARMA and MAM4. While emissions and removal for BC and primary organic aerosols are very similar between the different model configurations (Table 4), CARMA results in a smaller burden and lifetime compared to MAM4 due to differences in the removal of aerosol sizes. CAMchem CARMA and MAM4 are generally within the error bars of ATom observations except for the tropical Atlantic 610 upper troposphere and for the tropical Pacific for MAM4. In general, CARMA is closer to the observations than MAM4, which shows larger values than CARMA (Figure 15). Since both model configurations are based on the same wet removal scheme (Section 2.4.2.), differences are likely due to the different microphysical models with a more comprehensive description of the aerosol size distribution in CARMA.

Secondary organic aerosols (SOA) are based on the VBS approach in CAMchem, WACCM-MA uses a more simplified 615 approach assuming only one volatility bin. Differences in the SOA aerosol production and removal processes between CAMchem and WACCM-MA result in larger production and removal for WACCM-MA CARMA (Table 4). However, the different aerosol schemes, CARMA and MAM4, show fairly similar results in CAMchem in production and removal processes and compared to ATom observations. In contrast to both microphysical schemes in CAMchem, WACCM-MA MAM4 does not include photolysis in the standard version used here, while photolysis of SOA has only been added in WACCM-MA CARMA. In 620 addition, WACCM-MA MAM4 shows a significantly smaller production of SOA through gas-to-aerosol exchange and a much smaller total burden. Comparison of CAMchem between MAM4 and CARMA and ATom observations (Figure 16) shows an underestimation of OA at the surface and lower troposphere. A better agreement to observations occurs in the tropical mid-troposphere and SH mid-latitudes. The significant underestimation of OA, especially in the NH mid-to-high latitudes, is likely a result of the underestimation of SOA aerosol precursor emissions, including anthropogenic sources and biomass burning, 625 which is independent of the aerosol scheme since both aerosol models show the same bias.

### 5.2.3 Sulfate Aerosol and Extinction

The tropospheric sulfate burden in CARMA is 20 % larger for WACCM-MA and close to 40 % larger for CAMchem compared to MAM4. This is much closer than earlier reported differences of a factor of 2.8 between CESM1(CARMA) and MAM4 (Yu et al., 2015) (Table 4). The production of sulfate by aqueous-phase $SO_2$ oxidation and the production from $H_2SO_4$ 630 is slightly smaller in CARMA compared to MAM4, with larger values for CAMchem than WACCM-MA due to the more comprehensive chemistry and larger tropospheric ozone values in CAMchem. The chemical production of sulfate from $H_2SO_4$ is of similar magnitude and very similar among the different model configurations, with slightly larger values for CARMA compared to MAM4, leading to somewhat larger total production using MAM4 than CARMA (Table 4). As with the production, wet and dry removal is slightly smaller in CARMA than in MAM4. The stratospheric aerosol burden (the differences between



**Figure 15.** As Figure 13, but instead for black carbon.

the total and the tropospheric burden) is larger in MAM4 using CAMchem and similar to CARMA for WACCM-MA. Differences in burden can be a result of the details of the aerosol size distribution, which also leads to differences in stratospheric AOD (as discussed above).

Comparisons to ATom aircraft observations show very similar values for CAMchem CARMA and MAM4 at the surface, with MAM4 showing overall smaller values than CARMA throughout the altitude range, consistent with the smaller tropospheric aerosol burden in MAM4 (Figure 17). MAM4 tends to underestimate sulfates in the lower troposphere, while CARMA tends to overestimate sulfates in the mid-troposphere. Both models underestimate sulfate in the boundary layer in the SH mid-latitudes and other regions over the Pacific. A possible way to increase sulfates in this region is to consider that marine organics and sulfates are mixed with sea salt, as done in Yu et al. (2015), who assigned a fraction of sea salt flux to sulfate and marine organics.

The combined effect of aerosols on AOD can be identified by comparing aerosol extinction between the different configurations (Figure 18). MAM4 is within the error bars of the observations for most regions besides the tropical troposphere above





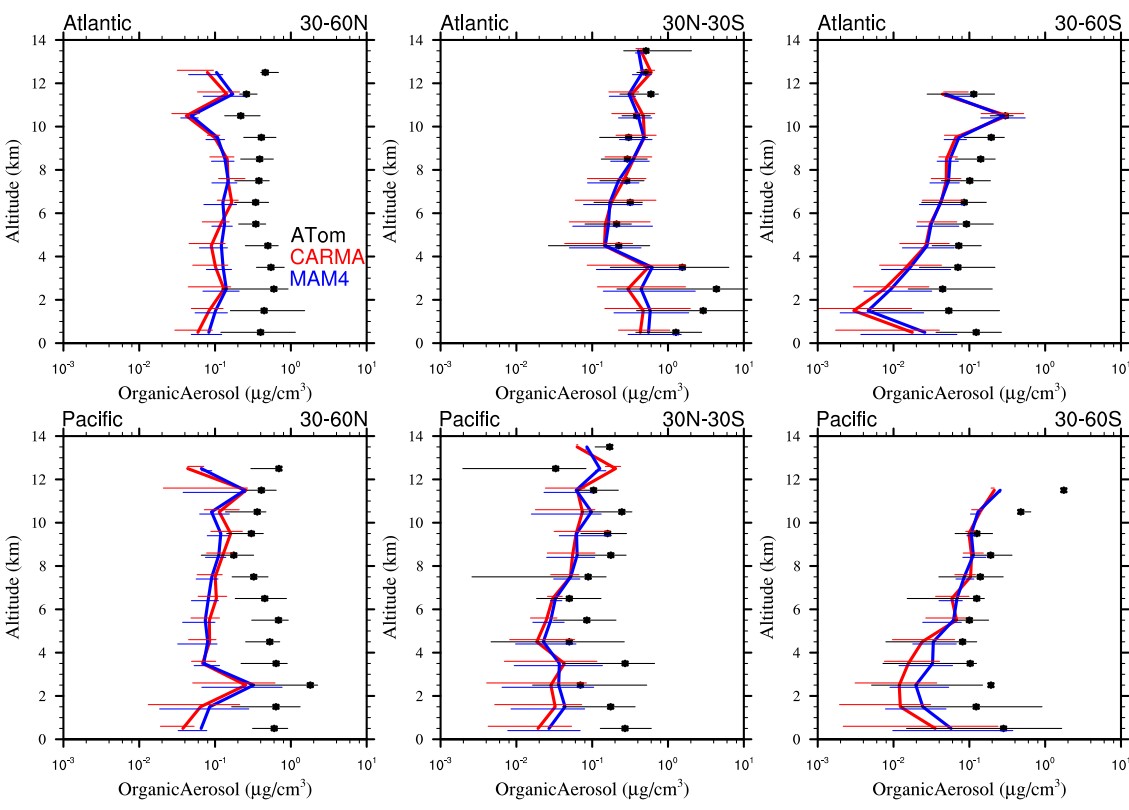

**Figure 16.** As Figure 13, but instead for primary plus secondary organic aerosols.

6 km over the Atlantic. CARMA reproduces the lower troposphere very well, while it shows an overestimation above about 6 km for most regions. This good agreement in CARMA at the surface is likely due to the fairly good representation of sea salt and dust. The overestimation in the free troposphere in CARMA may be the result of an overestimation of sulfates in this region and differences in the size distribution. However, ATom observations are still a snapshot in time, and there are strong vertical gradients in extinction. Global models cannot locate convection well, which may contribute to difficulty matching local data on extinction.

### 5.2.4 Aerosol size distributions

Number density distributions have been derived for distinct bins using observations from all four ATom campaigns (Brock et al., 2021). These are compared to the CARMA and MAM4 number density distribution over the Atlantic (Figure 19). The numbers of each of the different bins have been averaged for a large region (0-30°N, 0-30°S, 30-60°N) and for altitudes between 1 - 6 km (top row) and 6 - 12 km (bottom row, left and middle). We also compare stratospheric values, defined for all observations that show ozone > 150 ppb in the bottom right of Figure 19. CARMA size distributions are shown for pure sulfates (blue) and mixed aerosols (red) separately. Since these two CARMA aerosol groups cover different size ranges and sizes, we are not combining them here. The larger of the two groups in the logarithmic representation in Figure 19 can be viewed as the dominant group, which is comparable to the total number shown in the observations.



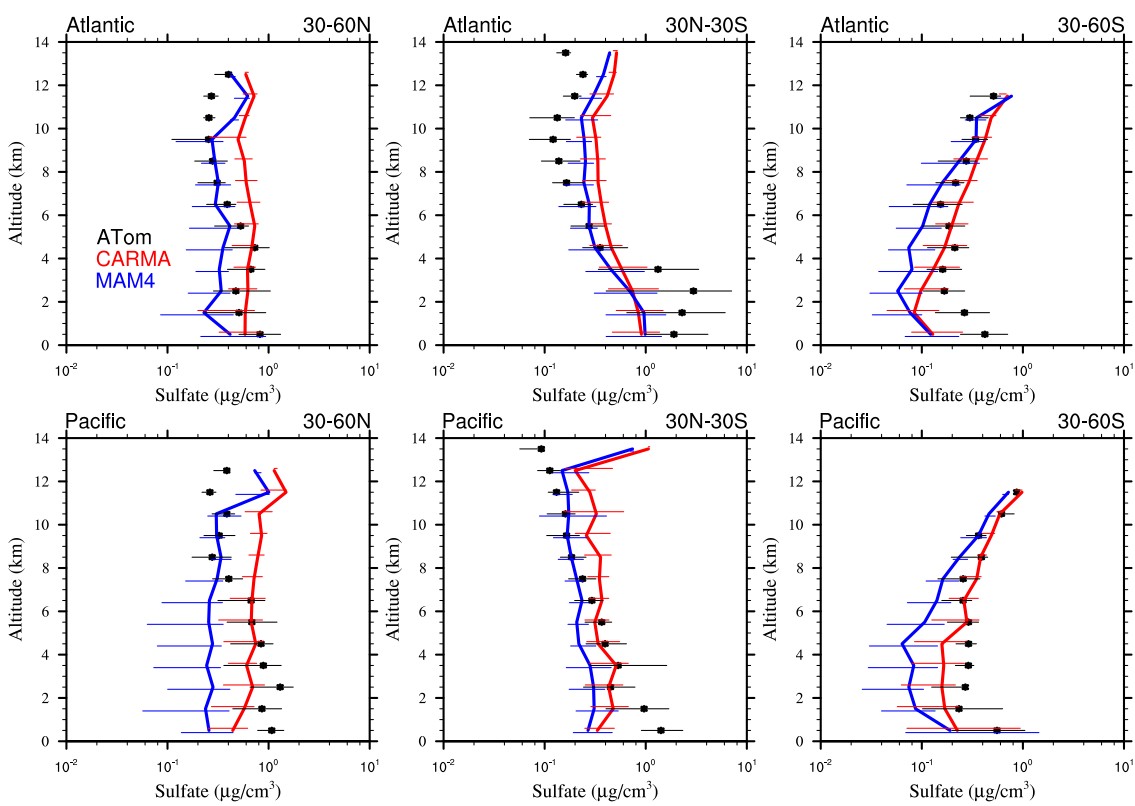

**Figure 17.** As Figure 13, but instead for sulfate aerosols.

For the Tropics in the lower troposphere averaged between 1-6 km (Figure 19, left top panel), both CARMA and MAM4 represent the shape of the accumulation mode aerosol distribution observed during ATom. The models also represent the second peak in the distribution between 1 and 6 $\mu$m for the tropical NH Atlantic, which is expected to be a result of the contribution of

dust from West Africa. Observations show very similar values for the larger sizes. While the MAM4 accumulation model also reproduces the shape of the ATom size distribution well and reproduces some of the larger sizes above 1 $\mu$m, the fairly narrow peak of the coarse mode cannot reproduce the largest sizes. There is also a gap between the Aitken and coarse mode around 1 $\mu$m that is not covering aerosols in that range. Furthermore, CARMA slightly underestimates Aitken-mode numbers, while MAM4 does not capture nucleation-mode sizes below 0.003 $\mu$m.

For 30-60°N over the Atlantic (similar to values over the Pacific and in the SH), the number of the mixed aerosols in CARMA is significantly higher than observed, with some overestimation also in MAM4. The mixed aerosol group in this CARMA setup is restricted to sizes from 0.05-8.7 $\mu$m. However, the lowest bin size is still too large to represent, for example, the nucleation of SOA. This results in an accumulation of newly formed SOA particles in the smallest mixed group bin, which is likely to lead to larger numbers than observed, in particular in the tropical mid-troposphere and polluted areas. The mixed group bin

structure in this model had been adopted from Yu et al. (2015). This comparison reveals the need for mixed groups that cover smaller aerosol sizes than in our model configuration. Still, the number of bins is an adjustable parameter and can be changed in the future. The pure sulfate aerosol group in CARMA covers much smaller bins than the mixed aerosol group and shows



**Figure 18.** As Figure 13, but instead for extinction.

an underestimation for aerosols between 0.005-0.1 $\mu$m in particle radius compared to observations. The lack of smaller size bins in the mixed aerosols is a likely reason for not representing these sizes correctly, as discussed above. The MAM4 Aitken

mode reproduces the number size distribution for sizes around 0.01 $\mu$m for the lower troposphere but overestimates numbers between 6-12km.

The shape and magnitude of the number distribution of aerosols in the lower stratosphere (over the Atlantic and Pacific) are represented well in CARMA compared to the ATom observations, Figure 19, bottom right panel. The model shows a slight underestimation between 0.01 and 0.1 $\mu$m. While there may be a lack of SOA in these size ranges, the underestimation may

also result from missing meteoric dust and nitrates in the model (Murphy et al., 2021). MAM4 overestimates Aitken and coarse mode numbers, in agreement with what was shown in Figure 9, compared to the Wyoming balloon observations.

Number density distributions in comparison to ATom observations are also derived for CAMchem CARMA using the Vehkamäki nucleation scheme (Figure A6). For the lower troposphere, pure sulfates show a significant underestimation in number, whereas the Zhao scheme produces values that are much closer to the observations. On the other hand, for the upper

**Figure 19.** Number density distributions of CAMchem CARMA results for size bins of the mixed aerosol (red) and pure sulfate (blue) group, and CAMchem MAM4 modes (green lines) to ATom1 to ATom 4 number size distributions from aircraft observations (black), averaged over three regions, 0-30° N, 0-30° S, and 30-60° N, and over 1 - 6 km altitude (top row), 6-12 km bottom left and middle panels, and for stratospheric air masses with ozone mixing ratios > 150ppb (bottom right). The model results were saved on the flight track using the closest location of each 1-minute data point and then averaged over the same regions as the observations.

troposphere and lower stratosphere, the Vehkamäki nucleation scheme does produce a reduced overestimate in the number of the very small bins of the pure aerosols group, as the case for the Zhao scheme. However, the numbers of sizes that included most of the mass (larger 0.01 $\mu$m) are still largely underestimated. We conclude that the Zhao nucleation scheme is more suited for reproducing aerosol distributions in the troposphere and lower stratosphere compared to the Vehkamäki nucleation scheme.

Comparisons between MAM4 and CARMA to ATom numbers in different modes are shown in Figures A7 – A9, based on

derived information by the ATom aerosol dataset for Aitken, accumulation, and coarse mode and compare those directly to MAM4. For CARMA, we derive numbers based on the same size ranges used for the data. These comparisons confirm earlier findings that CARMA underestimates Aitken mode aerosols, likely due to the setup of the mixed aerosol group, while MAM4





is generally within the error bars of the observations. On the other hand, both CARMA and MAM4 are generally within the error bars for the number in the accumulation and coarse mode according to ATom observations, in particular for the Tropics.
However, comparisons of mode averages do not reveal shortcomings in the size distributions, as discussed above.

## 6 Discussion

The performance of both microphysical models, CARMA and MAM4, has been tested in two different CESM2 configurations nudged to meteorological reanalyses, the CESM2 CAMchem model with a low top at 42 km and a horizontal resolution of 0.9° x 1.25° with comprehensive tropospheric and stratospheric chemistry, and the CESM2 WACCM-MA version with a model top
at 140 km, a horizontal resolution of 1.9° x 2.5° with middle atmosphere chemistry. To evaluate the evolution of aerosols and aerosol properties after the Mt Pinatubo eruption (the largest volcanic eruption in the last 25 years), we performed different sensitivity simulations between 1990 and 1995 with WACCM-MA using both CARMA and MAM4. We also evaluated tropospheric aerosol properties between 2000 and 2020 and between 2016 and 2018 in comparison to observations from the ATom aircraft campaign.

Sensitivity simulations with WACCM-MA compared to the GloSSAC SOAD climatology reveal the importance of applying sulfur injections over a larger region for simulating the Mt Pinatubo eruption using a global model to be able to reproduce the observed evolution of aerosols. Injections in a single column on a grid cell in horizontal dimension do not result in the observed spread of aerosols into both hemispheres after a few months of the eruption. The coarse model resolution of the global model cannot reproduce details of the specific meteorological situation during the time of the eruption and cannot
reproduce fine-scale dynamics. A regional injection better reproduces the initial spread and improves the aerosol movement later compared to observations. Furthermore, while the chemistry and physics in WACCM-MA are the same in MAM4 and CARMA, results from different aerosol models show significant differences in the initial aerosol formation (size, surface area density, and composition). Reasons for these differences lie mostly in the details of the parameterization of the size distribution of aerosols. The modal approach in MAM4 (which does not separate between nucleation and Aitken modes) shows a much
faster formation and growth of sulfate aerosols after the eruption compared to the sectional model approach using CARMA, where nucleated particles first accumulate in a much smaller size bin than the MAM4 Aitken mode, restricting coagulation and initial growth.

However, after a few months, aerosol particles in CARMA grow larger than in MAM4. Comparisons with balloon observations indicate a more reasonable representation of the aerosol size distribution following the Mt Pinatubo eruption in CARMA
than in MAM4. CARMA and MAM4 both overestimate the number of the large 1 $\mu$m particles. CARMA agrees with the accumulated number from the direct measurements for the smaller size bins, while MAM4 underestimates the number between 0.03 and 0.4 $\mu$m and overestimates the Aitken model number. Comparisons of the volume size distribution between CARMA and MAM4 reveal that most of the mass after a few months is in the bins that describe the coarse mode, which is well represented in CARMA. However, in MAM4, the coarse mode distribution is narrower than the corresponding size distribution in
CARMA and the observations. The smaller coarse mode mass and larger accumulation mode mass in MAM4 are aligned with





the smaller effective radius, resulting in less aerosol removal and, therefore more total mass in MAM4 after the Mt Pinatubo eruption. Furthermore, some overestimation of the large 1 $\mu$m particles in the largest pure sulfate group in CARMA indicates that the chosen range of the pure sulfates may not cover large enough bins to allow the aerosols to grow larger and instead traps the particles in the largest bin. This limitation may have implications for representing large injections; for example, if climate

intervention in the form of stratospheric aerosol injection is applied, implementing larger-size bins should be considered in this case.

While the results of each of the models can, to some degree, be adjusted depending on the injection amount and region, this comparison reveals large uncertainties in our ability to simulate the initial formation of aerosols in a volcanic plume. In addition to the resolution issue, other processes are still not included in either model. For example, the injection of volcanic ash, which

is expected to result in a faster formation and removal of injection sulfur by heterogeneous reactions on rock surfaces, has not been included in the current model (Zhu et al., 2020). Furthermore, the lack of observations during the initial volcanic eruption makes it harder to estimate the exact injection amount of gases and aerosols. The current uncertainty can have implications for simulating the effects of stratospheric aerosol injections. Some differences between MAM4 and CARMA are the result of using two different nucleation schemes (the Zhao scheme in CARMA and the Vehkamäki scheme in MAM4). Applying the

Vehkamäki nucleation scheme in CARMA increases the initial nucleation rate after the Mt Pinatubo eruption and results in a build-up of more mass in the stratosphere. However, comparisons of the Vehkamäki nucleation scheme for the troposphere reveals reduced performance and we recommend using the Zhao scheme for applications in the troposphere and UTLS.

Considering the background stratospheric aerosol optical depth (SAOD) and changes of smaller eruptions after the year 2000, both WACCM-MA and CAMchem using CARMA and MAM4 reproduce the observations within the error bars, with

some overestimation of SAOD for small eruptions. CARMA agrees with the stratospheric number density size distribution from balloon observations, while MAM4 overestimates the number of aerosols in large and small sizes. Comparisons to AOD from the MODIS satellite data reveal regional differences in the tropospheric aerosol representation in MAM4 and CARMA. In general, MAM4 overestimates several regions that are in particular impacted by dust and sea-salt emissions and in particular, shows a significant overestimation of AOD in the SH and an underestimation in the NH for June-July-August. CARMA more

generally underestimates AOD over the ocean. Differences in the emissions and dry removal processes of dust and sea salt are obvious in comparing the global burdens of the different model configurations. Future work needs to include aligning these different approaches to improve comparisons and test the inclusion of marine organic aerosols (e.g., Zhao et al., 2021). CARMA more comprehensively represents larger bins than MAM4 compared to observations, leading to different removal rates in the two aerosol models.

Comparison to ATom aircraft observations over the remote regions over the Atlantic and Pacific show both aerosol schemes are generally within the error bars using CAMchem. CARMA shows a better representation of black carbon than MAM4. However, both aerosol models overestimate BC over the Atlantic in the Tropics. In general, larger sulfate mixing ratios are simulated using CARMA. However, both models underestimate surface mixing ratios in the Tropics and the SH. CARMA and MAM4 reproduce observed extinction at the surface; however, CARMA overestimates aerosol extinction in the mid-

troposphere, which may be due to the larger sulfate values in that region.





Considering aerosol size distributions reveal that CARMA underestimates bin sizes between 0.01-0.04 $\mu$m, especially in the upper tropical troposphere where the largest formation of SOA is expected. This is likely because the mixed aerosol group in CARMA covers sizes between 0.05 and 8.7 $\mu$m, which may not be small enough to represent the nucleation process for SOA. Additionally, the current CARMA model does not include meteoric dust, which has also been suggested to increase the

number of aerosols in these sizes in the stratosphere (Murphy et al., 2021). On the other hand, the narrow coarse mode width in MAM4 underestimates observed coarse mode numbers and results in reduced aerosol deposition, as pointed out in earlier studies. Other shortcomings in the current simulations (MAM4 and CARMA) include the missing or insufficient inclusion of nitrate aerosols in the models. A sectional nitrate model has been recently implemented in CESM1-CARMA (Yu et al., 2022) and in CESM2 MAM4 (Jo et al., 2021).

This study focuses on the evaluation of aerosols and aerosol properties in CARMA and MAM4 using model configurations that are nudged to meteorological reanalyses. Future work will focus on developing a configuration that is coupled to the ocean and ice using CARMA. This will require a more detailed investigation of the performance of clouds and the resulting effects on radiative forcing, which is beyond the scope of this study.

## 7   Conclusions

The implementation of CARMA in CESM2 configurations (CAMchem and WACCM-MA) allows for the first time a comparison of a modal and a sectional aerosol model in the same CESM2 configurations. Advantages and shortcomings can be identified in testbed experiments while nudging meteorological conditions, e.g., wind and temperature fields, to meteorological reanalyses. With this configuration, differences in the formation of aerosols in the volcanic plume after the Mt Pinatubo eruption help to identify specifics of the different aerosol microphysical schemes. Additional comparisons to observations of

other and future volcanic eruptions will allow a better understanding of the performance of the different aerosol schemes in the model. Detailed comparisons to aircraft observations identify shortcomings dependent or independent of the specific aerosol model, allowing suggestions for more targeted model improvements in future work. Further applications will be towards a fully coupled model version that will allow the quantification and, eventually, reduction of uncertainties in aerosol interactions with the climate system for climate-relevant studies.

*Code and data availability.*    The code of the model used in this work is available through Zenodo: https://doi.org/10.5281/zenodo.7829697 and Github: git clone -b carma-trop-strat12 https://github.com/fvitt/CAM.git camroot. Model results used in this study are available for download by request to the authors. The Wyoming balloon data used in this study are available in the data repository at the University of Wyoming Libraries: Deshler, T., and Kalnajs, L. E. (2022). University of Wyoming stratospheric aerosol measurements | Mid latitudes [Dataset]. https://doi.org/10.15786/21534894. MODIS data used for this study are available at MOD08-M3.061: MODIS/Terra Aerosol Cloud Water

Vapor Ozone Monthly L3 Global 1Deg AOD 550 dark target deep blue combined mean https://ladsweb.modaps.eosdis.nasa.gov/archive/allData/61/MOD08 M3, DOI:10.5067/MODIS/MOD08-M3.061.



*Author contributions.* ST, MM, YZ, FV, CB, PY: developed and implemented the model. ST, MM, YZ performed model simulations and analyzed the results. DF produced comparison plots with MODIS data. TD helped with analyzing Wyoming balloon observations. ST led the writing of the manuscript with valuable comments and discussions by all co-authors

*Competing interests.* At least one of the (co-)authors is a member of the editorial board of Geoscientific Model Development.

*Acknowledgements.* We thank Daniele Visioni and Louisa Emmons for helpful comments and suggestions. This project received funding from NOAA Climate Program Office Earth's Radiation Budget Awards Number 03-01-07-001 and NA22OAR4310477. The Wyoming balloon measurements were completed under support from the National Science Foundation. The CESM project is supported primarily by the National Science Foundation. This material is based upon work supported by the National Center for Atmospheric Research, which is

a major facility sponsored by the NSF under Cooperative Agreement No. 1852977. Computing and data storage resources, including the Cheyenne supercomputer (doi:10.5065/D6RX99HX), were provided by the Computational and Information Systems Laboratory (CISL) at NCAR.



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



**Figure A1.** Timeseries of airmasses in the volcanic plume between 30°S and 30°N and between 200 and 5 hPa, defined by grid points with stratospheric surface area density larger than 0.1 $\mu$m using daily averaged model output for different chemistry and aerosol variables, over the first 30 days (first and second row) and over the first six and half months (bottom row), comparing WACCM-MA experiments comparing different nucleation schemes and injection amounts using CARMA and MAM4.

## Appendix A: Supporting Figures

This section includes supporting material in the form of additional Figures, as referred to in the main text.





**Figure A2.** Size distribution comparisons of different model experiments using CARMA compared to Wyoming balloon observations for October 1991 and March 1992 after the eruption of Mt Pinatubo. Model experiments used regional injections of 7 TgS and the Zhao nucleation scheme (top) and the Vehkamäki nucleation scheme (bottom).





**Figure A3.** As Figure 10, but for tropospheric ozone column (in DU) (for ozone > 150 ppb in the model)





**Figure A4.** As Figure 13, but instead for ozone.





**Figure A5.** As Figure 13, but instead for carbon monoxide.



**Figure A6.** Number density distributions of CAMchem CARMA results using the Vehkamäki aerosol nucleation scheme (instead of the Zhao nucleation scheme shown in Figure 19),for size bins of the mixed aerosol (red) and pure sulfate (blue) group, and CAMchem MAM4 modes (green lines) to ATom1 to ATom 4 number size distributions from aircraft observations (black), averaged over three regions, 0-30°N, 0-30°S, and 30-60°N, and over 1 - 6 km altitude (top row), 6-12 km bottom left and middle panels, and for stratospheric air masses with ozone mixing ratios > 150ppb (bottom right). The model results were saved on the flight track using the closest location of each 1-minute data point and then averaged over the same regions as the observations.



**Figure A7.** As Figure 13, but instead for Aitken mode number.





**Figure A8.** As Figure 13, but instead for accumulation mode number.





**Figure A9.** As Figure 13, but instead for coarse mode number.