# Peer review of "Description and performance of the CARMA sectional aerosol microphysical model in CESM2"

_Geoscientific Model Development, 2023_

## Author Comment (AC1)

**Response to the reviewers.**

We thank the reviewers for their helpful comments and suggestions and hope that we addressed all the changes sufficiently to be considered for publication. The responses to the reviewer's comments are shown in blue.

**Reviewer 1**

The paper provides a comparative evaluation of the performance of two aerosol microphysical schemes in simulations performed with two different configurations of the Community Earth System Model (CESM2). In one the sectional CARMA module is compared to the modal MAM4 module in the high-top version of CESM2 called WACCM-MA and applied to study of the 1991 Mount Pinatubo volcanic eruption and subsequent period. In the other the two aerosol modules are run in the low-top CAMchem configuration and applied to simulations of tropospheric and lower stratospheric aerosol in the context of the 2016-2018 ATom airborne observations. It is shown that for Pinatubo the performance of both CARMA and MAM4 are improved by distributing emissions over a region instead of prescribing over the volcano column alone. CARMA results in overall larger particles following Pinatubo, with a correspondingly shorter atmospheric lifetime. Two different nucleation schemes are tested in CARMA, its default Zhao scheme and the Vehkamäki scheme used in MAM4. Overall CARMA's performance is better with its default scheme. Tropospheric aerosols are compared to MODIS AOD and ATom profile observations. Modeled profiles are generally similar between CARMA and MAM, each with its strengths and weaknesses in terms of species and regions. CARMA somewhat overestimates the mid-tropospheric sulfate relative to ATom, and MAM has consequently better agreement in the extinction profile. CARMA is shown to have overall better agreement with observed particle size distributions for both Pinatubo and ATom cases.

The paper is well organized and relatively comprehensive. It is suitable for publication with minor revisions. I suggest a couple of points below that could be expanded on to improve the discussion, and suggest numerous minor points.

Major points

Line 183 refers to sub-stepping for stability in process rates. Some additional information would be useful here, particularly as to how it impacts performance and accuracy. Do MAM4 and CARMA take similar approaches? Do they have similar accuracy tolerances? What are those?

CARMA and MAM4 have different approaches when it comes to microphysical calculations. MAM4 does not include any sub-stepping, while CARMA does; see updated text below, which also addresses questions and performance and accuracy.

To address the reviewers' comments, we changed the text and included additional information:

"Aerosol microphysics for CARMA (Process 6) is done last, which includes sedimentation, dry deposition, molecular diffusion, and coagulation, followed by nucleation, growth, and evaporation. The calculation of nucleation, growth, and evaporation rates are performed simultaneously and undergo convergence checks to make sure that the gas concentration (here H2SO4) is not negative, and that temperature and supersaturation changes do not exceed pre-defined thresholds. If convergence cannot be reached due to large process rates, the model will retry with shorter sub-steps until convergence is reached. This will result in a stable solution for each modeled time step. The sub-stepping will add some processing time to the model, particularly during a spin-up phase when gases and aerosols have not reached sufficient balance."

Section 2.4.5 could be more explicit with respect to the size distributions of emissions, specifically how they compare to each other between CARMA and MAM. Lifetime numbers reported in Table 4 are more typical of AeroCom models in MAM than they are for CARMA, which is remarkably short for both dust and sea salt. This could be a consequence of emissions being strongly weighted toward largest size bins in CARMA. Some further detail here would be helpful instead of just referring to previous work. The discussion in 5.2.1 expands a bit more on this as well. The discussion seems a bit simplistic. It would be better to present in terms of efficiencies of processes instead of references to burdens. I know there is non-linearity in the microphysics, but in terms of the sink processes they are quasi linear in terms of the burden, so it's not very illuminating to frame it that way. Instead, what seems clear, is that CARMA prefers dry deposition, which necessarily leaves less aerosol for wet removal. MAM is more the opposite.

We agree with the reviewers' observation that the lifetimes of aerosols are much shorter, particularly for dust and sea salt, in contrast to most AeroCom models. Lian et al. (2022) compared the global dust size distribution of an earlier version of CARMA to AeroCom models (Adebiyi and Kock, 2020). They showed that CARMA does agree very well with observations, while most AeroCom models show too large numbers of smaller sizes and too small numbers of large sizes (see Figure below). The bin representation of the size distribution in CARMA moves aerosol emissions into large size-bins, which then results in a fast deposition of these larger particles. This, in general, leads to larger emission and dry deposition rates in CARMA compared to MAM4, where the size distribution of the mode constrains the emissions. For MAM4, the coarse mode with a sigma value of 1.2 results in a narrower distribution of emissions and mass. In CARMA, due to having emissions in larger size bins and a resulting larger dry deposition, a smaller fraction of the aerosol mass is reaching higher altitudes, resulting in smaller wet removal and a smaller aerosol burden in CARMA compared to MAM4.

Following the reviewers' advice, we are adjusting Section 2.4.5 and the discussion in 5.2.1:

Section 2.4.5: We add the following text at the end of the second paragraph:

"CARMA and MAM4 emissions are calculated as mass emission fluxes and are distributed over all the mass bins or modes, respectively. CARMA emits increasingly more mass into larger bins for sea

salt and dust. This results in the relatively large total emissions in CARMA and consequently larger dry deposition due to larger deposition rates of larger particles, as discussed below (Section 5.2.1)."

[Figure]

**Figure 1.** Comparison of the simulated and measured normalized global mean dust size distributions. The dust mass size distributions are divided by the total dust mass integrated over the size range (i.e., the area under each d$M$ / dln$D$ curve). The global and annual mean dust size distribution simulated by CESM1/CARMA with the dust emission parameterization, described in P. F. Yu et al. (2015), is shown by the dashed red line. The simulation by CESM1/CARMA with the modified emission parameterization is shown by the solid red line. Temporal variabilities (1 standard deviation) from P. F. Yu et al. (2015) and CESM1/CARMA are denoted by green and cyan lines. The simulated size distribution by the AeroCom models reported in Adebiyi and Kok (2020) is denoted by the gray lines. The measured dust size distribution derived from the global measurements reported in Adebiyi and Kok (2020) is denoted by the solid black line. The shading represent the 95 % confidence interval.

We further changed the text in Section 5.2.1:

"Differences in sea-salt and dust burdens between CARMA and MAM4 are the result of differences in how emissions are calculated, the microphysical parameterizations that result in different aerosol size distributions, and resulting differences in the removal processes. In particular, sea salt and dust burdens derived from the CARMA are only about half the amount derived using MAM4. The two to three times larger emissions in CARMA compared to MAM4 are the result of how emission fluxes are distributed into the bins and modes, with more mass being emitted in the larger size bins in CARMA. On the other hand, coarse mode emissions in MAM4 are smaller in total mass than in CARMA since they are constrained by the narrow standard deviation of 1.2, which was chosen to accommodate the stratospheric coarse model sulfate (Mills et al., 2016, Niemeier et al., 2011). The emissions of larger particles in CARMA also lead to the large deposition of aerosols that have a larger fall velocity than smaller-sized aerosols, as discussed by Yu et al. (2015). On the other hand, MAM4 shows a larger number of coarse model sizes above the boundary layer (1-6km) for regions with a larger dust and sea-salt occurrence (as discussed below in Section 5.2.4), resulting in relatively more wet removal than CARMA. The larger burden in MAM4 and much smaller dry removal results in a much longer lifetime than in CARMA. While most of the AeroCom model shows similar

lifetimes to MAM4 (Adebiyi and Kock, 2020), Lian et al. (2022) have demonstrated that the CARMA aerosol size distributions agree much better with observations, in particular in reproducing larger numbers for larger size bins and a smaller number for the smaller size bins.".

Minor points

Line 102: For completeness, what is the mode width for the primary carbon mode?

We added the information on the primary carbon mode to the text:

"The geometric standard deviation in MAM4 for the different modes is Aitken: 1.6, accumulation: 1.6, **primary carbon mode: 1.6**, and coarse:1.2. (Liu et al., 2016; Mills et al., 2016)."

Line 156: "Therefore the wet radius below 190 K" is a sentence fragment of unclear intent.

The text was messed up. We change:

"While Yu et al. (2015) assumed no particle swelling below 190 K, in this study, we use the relative humidity at 190 K to calculate the particle swelling. Therefore the wet radius below 190 K. "

To

While Yu et al. (2015) assumed no particle swelling below 190 K, in this study, we use the relative humidity at 190 K to calculate the particle swelling and the wet radius below 190 K.

Line 169: s/b McFarlane

Thanks for the note; we have changed that.

Line 171: "two moment cloud microphysics (?)" Unclear what the (?) means, maybe a missing reference?

Yes, we were missing a reference; the correct sentence is:

"These are coupled to aerosol activation (Abdul-Razzak and Ghan, 2000), eddy diffusion (Process 1 in Figure 1), two-moment cloud microphysics (Gettelman and Morrison, 2015), and convective and stratiform wet removal (Process 2 in Figure 1). "

Line 209: s/b McFarlane

We fixed this.

Line 255: Here and elsewhere this note about photolysis of SOA. What does this mean? Is this a destruction pathway, or something to with photolysis chemistry? A reference would help.

We refer the reviewer to a previous sentence: "We also include SOA photolysis, assuming a reaction rate that is 0.04 times the photolysis rate of nitrogen dioxide, as discussed in Hodzic et al. (2015), and add the SOA formation from glyoxal in aqueous aerosols (Knote et al., 2014) as also done for MAM4 in CAMchem.". To clarify this some more, we changed the sentence to:

"We also include SOA photolysis **as a sink of SOA in the upper troposphere**, assuming a reaction rate that is 0.04 times the photolysis rate of nitrogen dioxide, as discussed in Hodzic et al. (2015), and add the SOA formation from glyoxal in aqueous aerosols (Knote et al., 2014) as also done for MAM4 in CAMchem."

Line 284: Suggest "correspondingly" instead of "somewhat" smaller throughput.

We changed the sentence to: with a smaller throughput for CARMA in the specific configuration"; therefore, removed "somewhat".

Line 323: Do you really mean QFED here? Citation is Darmenov and da Silva (2015, https://ntrs.nasa.gov/citations/20180005253)

Yes, we use QFED emissions for biomass burning, but use emission factors from the FINN inventory. We add the appropriate reference for QFED here:

"For the period between 2001 and 2020, we use CAMSv5.1 anthropogenic emissions and biomass burning emissions derived from QFED $CO_2$ fields **(Darmenov and da Silva (2015)**, multiplied by the species emissions factors collated in Fire INventory from NCAR Version 1.5 (FINNv1.5; Table S1 at http://bai.acom.ucar.edu/Data/fire/; Wiedinmyer et al., 2011).

Figure 5, second panel: Since the behavior essentially asymptotes at 30 days, suggest you compress x-axis range to allow better inspection of the first 10 days in this figure, where the models are quite different.

We chose the range of the x-axis to compare both the initial change in $SO_2$ e-folding time and the longer-term effect on the $SO_2$ burden, and therefore prefer to keep the range as shown. We think that the figure does clearly show differences in the first 10 days.

Line 485 and rest of paragraph: Figure 9 is stated but actually Figure 8 is meant.

Line 502: Figure 9 is meant instead of Figure 8.

We agree with the reviewer and fixed the reference to the figures

Table 4: Caption states lifetime given in years. This is incorrect, it should be days.

We agreed and changed the caption to lifetimes in days instead of years. In addition, we updated the number for SOA lifetime because we failed to include all the sinks for SOA: wet and dry removal and photolysis rates.

Line 537: What MODIS products are used here?

Please see line 523 in the manuscript text, where we have detailed the description of the MODIS product used.

Figure 11: Positional labeling in caption caption (top, middle, bottom…) would make sense if there are six panels, but there are eight. Suggest label a, b, c, …

We agree that the labeling was not ideal. We changed the figure caption to:

Aerosol Optical Depth in the visible (550nm), June, July, and August (JJA) averaged between 2001 and 2020 from MODIS (TERRA) observations (first row left), MERRA (first row right), and from CAMchem CARMA (second row left) and CAMchem MAM4 (second row right). The third and fourth rows show absolute (third row) and relative (fourth row) differences between CAMchem CARMA and the MODIS observations (left) and between CAMchem (MAM4) and observations (right).

Line 576: "settling" instead of "settlement"

changed

Line 582: Here and rest of paragraph Figure 11 actually s/b Figure 13 and Figure 12 actually s/b Figure 14
We thank the reviewer for pointing this out; we fixed the references to the figures.

**Reviewer 2**
The manuscript "Description and performance of the CARMA sectional aerosol microphysical model in CESM2" provides a comprehensive evaluation of the CARMA sectional model for stratospheric and tropospheric conditions comparing the model to the modal model MAM4 as well as observations. The model performance for describing stratospheric aerosol properties was evaluated by simulating the massive volcanic eruption of Mt. Pinatubo. The choice of applying the model for such an extreme case is appropriate as simulating aerosol properties in such conditions will be strongly affected by the numerical methods chosen for solving aerosol microphysical processes. For example, modal and sectional models will result in very different simulated aerosol size distributions and thus radiative properties for aerosol populations. The paper is well written, extremely comprehensive, scientifically sound and I can recommend publishing it after the following minor issues have been addressed:

- As an overall comment, using the same global model host model for using both a model and a sectional model is a good way to compare the two aerosol models in the sense that the differences

would originate only from the description aerosol microphyscs. However, it is unfortunate that in these simulations, MAM4 and CARMA implementations had also quite large differences in, for example, online calculated emissions of sea salt and dust which result in very different amounts of emitted aerosol. Such big changes make it difficult to get a good picture on how much MAM4 and CARMA themselves contribute to the differences between the runs.

We agree that the differences in sea salt and dust are not a result of the different aerosol microphysical schemes and differences exist. This cannot be changed until sea salt and dust emission schemes are unified. However, various other comparisons between the different microphysical schemes were performed in this work, including stratospheric aerosols and size distributions and comparisons of other tracers than dust and sea salt.

Specific comments:

Page 4, Line 116: "For the stratosphere, sulfates are in equilibrium with the water, and a weight percent of H2SO4 is calculated based on the parameterization by Tabazadeh et al. (1997)." This is a very difficult sentence to understand. Do you want to say that for the stratosphere the water uptake is calculated using the parameterization by Tabazadeh et al. (1997)?

That is correct. We changed the sentence for clarification:

"For the stratosphere, sulfates are in equilibrium with the water. The water uptake (and therefore the weight percentage of H2SO4) is calculated based on the parameterization by Tabazadeh et al. (1997)."

Page 4, Line 103: The choice for such a small geometric standard of 1.2 is not well explained in the manuscript. Later in the manuscript, it is briefly mentioned that it is required to be able to simulate stratospheric aerosol from strong volcano eruptions (Niemeier et al., 2011). This should be explained already here.

Following the suggestion of the first and second reviewers, we changed this part to:

The geometric standard deviation in MAM4 for the different modes is Aitken: 1.6, accumulation: 1.6, primary carbon mode: 1.6, and coarse:1.2. The relatively small sigma value of 1.2 for the coarse mode had been chosen to accommodate the stratospheric coarse model sulfate (Mills et al., 2016) following Niemeier et al. (2011).

Page 5, Line 125 it is said: "sectional nitrate and ammonium (Yu et al., 2022), are not included in the current version of the model". Does this mean nitrate and ammonium are treated as bulk compounds of they are fully omitted?

It means that nitrates are completely omitted in CARMA.

Page 5, Line 132-133: "Currently, CARMA only allows one component of a group to be volatile. The addition of SOA in this model requires calculating SOA volatility (gas to aerosol exchange) in CAM." I don't understand these sentences; why is it required to calculate SOA volatility in CAM?

The current version of CARMA only allows one element in each group (in this case, sulfate is the so-called "concentration element" of the mixed aerosol group) to be volatile and experience evaporation or condensation. In our setup, both sulfate and SOA are volatile and experience gas-aerosol exchange. In order to simulate this in the model, we are imposing exchange rates of different volatility bins (following the VBS scheme) to the CARMA model that are not calculated within CARMA but rather within the chemistry of the atmospheric model CAM (outside the microphysical scheme). A more internal approach would include the gas-to-aerosol exchange for both SO4 and SOA in the microphysical model code and lead to somewhat more consistency. We are planning to develop the code in the future to add this option in the future. To clarify this in the text, we change it to:

"Currently, CARMA only allows one component of a group to be volatile, which is sulfate for the mixed and pure aerosol groups. The volatility of SOA (gas-to-aerosol exchange) is therefore calculated in the chemistry module in CAM, and the resulting rates are passed into CARMA."

Page 5, Line 148-149: "The model does not currently employ nucleation influenced by ammonia or organics, which is likely important near the ground." Please add a reference.

To further clarify, we change the sentence to:

"The model does not currently employ nucleation influenced by ammonia or organics, which is important for radiation and other aerosol processes (Lu et al., 2021)."

Page 6, Line 156: "Therefore the wet radius below 190 K." There is something missing from this sentence.

The text was messed up. We change:

"While Yu et al. (2015) assumed no particle swelling below 190 K, in this study, we use the relative humidity at 190 K to calculate the particle swelling. Therefore the wet radius below 190 K. "

To

While Yu et al. (2015) assumed no particle swelling below 190 K, in this study, we use the relative humidity at 190 K to calculate the particle swelling and the wet radius below 190 K.

Page 7, Line 170: Boundary layer is not a processes

We changed this to "planetary boundary layer processes, ..."

Page 8, Line 200: "It is based on the number of dust and sulfate" Should this be "amount" instead of "number"?

We do not agree to changing this to "amount". Here we mean "the number of aerosol particles". To clarify, we changed the sentence to: "It is based on the particle number of dust and sulfate within the mixed and the pure sulfate group in CARMA, considering only aerosols that are > 0.1 microns."

Page 9, Line 240-241: How is cloud-produced sulfate distributed between CARMA bins?

To address this comment, we changed the text to:

"Aqueous-phase reactions include reactions of aqueous sulfur by ozone and hydrogen peroxide to form SO4 and therefore depend on tropospheric chemistry. The produced sulfate is added into the cloud-borne aerosol MAM4 sulfate modes or CARMA bins proportional to sulfur mass in each bin."

Page 16: "Therefore, the acid molecules are rapidly lost from the gas phase and nucleate much faster." What does this mean? If sulfuric acid molecules are lost, why do they nucleate faster? I would expect a decrease in gas phase concentration to supress nucleation.

We realize the wording in the paper was misleading, and we changed it to:

"Therefore, the acid molecules more rapidly nucleate and further increase coagulation."

Section 5.1. when comparing modeled AOD to satellite AOD, was the model data collocated to satellite observations?

We have not co-located the satellite observations and the model data since we are using a 19-year average of AOD for both observations and the model. We assume that this would not result in a significant difference.

Page 27: It is speculated that MAM4 overestimates AOD in the southern hemisphere due to too strong sea salt emissions and both models overestimate AOD over South America due to secondary organic aerosol. In addition, some of the differences between the models have been suggested to come from different dust and sea salt emissions. Could this be diagnosed from the model output by breaking the AOD to components, i.e. what fraction of AOD comes from which compound? In addition, aerosol lifetimes are very different between the models and thus dry and wet removal might be even a more significant cause for intermodel differences.

We agree that breaking the AOD into different components would help to clarify and support the speculations. Unfortunately, the model simulations did not include AOD variables for the different components, and we cannot analyze this without rerunning the entire set of simulations. We are adding this option in future studies. We also agree that the aerosol lifetime is very different. We updated the text regarding the differences in lifetimes for dust and sea salt following suggestions

from reviewer 1 (see above). Indeed differences in dust emissions, which are more strongly emitted into larger bins, do cause stronger dry deposition, which further results in a smaller aerosol burden and wet removal.

Page 31, Line 600-601: "Differences compared to the observations in the tropical Atlantic mid-troposphere may be related to how CARMA and MAM4 apply wet removal.". Can you elaborate this?

CARMA and MAM4 include the same wet removal schemes, as discussed in Section 2.4.2, stating that both aerosol schemes use the same parameterizations. We agree that the sentence is misleading. For the SH, differences between CARMA and MAM4 are most likely the result of much larger aerosol burdens (and AOD) over Australia and South Africa in MAM4, which is also indicated by the overestimation of AOD compared to MODIS and MERRA. For the tropical Atlantic, both aerosol models overestimate dust. This could be a result of the details of the wet removal scheme, e.g., assumed hygroscopicity, or also potentially a lack of resolving convection in the model. We changed the text to:

"The main difference between CARMA and MAM4 is an overestimation of dust in MAM4 by one order of magnitude in the SH mid-latitudes, while CARMA agrees well with the observations. The derived AOD in CARMA over Australia and South Africa is represented better compared to observations than in MAM4. Differences between CARMA and MAM4 and the observations in the tropical Atlantic mid-troposphere may be related to the wet removal parameterization or shortcomings in resolving deep convection in the model. "

Page 33, Lines 635-637: "Differences in burden can be a result of the details of the aerosol size distribution, which also leads to differences in stratospheric AOD (as discussed above)." This sentence is ambiguous. Do you mean that the representation of the aerosol size distributuin leads to differences in burdens?

Yes. Using a bin model, with a more detailed resolved size distribution than a modal model, leads to differences in how mass is distributed between different aerosol sizes. These differences can lead to differences in dry and wet deposition, as discussed above (Reviewer 1) for dust and sea salt, and therefore impact the aerosol lifetime. To clarify, we update the sentence to:

"Differences in burden can result from the details of the aerosol microphysical description (using bins vs. modes), which results in differences in the aerosol size distribution and burden, due to differences in dry and wet deposition. This can also lead to differences in stratospheric AOD (as discussed above)."

Section 5.2.4: The comparison of size distributions seems a bit unfair to MAM4 since it uses Vehkamäki nucleation scheme which produces a negligible amount of nucleation in the troposphere. Was there a practical reason for not using Zhao et al. nucleation scheme (or any boundary layer

nucleation scheme) in MAM4? With this in mind, to have a better simultaneous comparison between observations and the two models, it would make sense to switch places between Figure 19 and A6.

The default nucleation scheme in MAM4 is the Vehkamäki nucleation scheme. The goal of this study was not to modify MAM4. We would like to keep Figure 19 as it is since we want to compare the default nucleation schemes used in each of the two aerosol models.

Page 40, Line 777: Please change "This will require a more detailed investigation of the performance of clouds" to something like "This will require a more detailed investigation of the performance of the model to simulate clouds"

Thanks for the suggestion; we have changed it accordingly.

Technical comments:

- Page 7, Line 171, reference missing

Thanks for pointing this out. We have added the missing reference.

- Figure 9 is discussed before Figure 8

There was a mistake in the order of the Figures; we have fixed that.

Niemeier, U., Schmidt, H. and Timmreck, C. (2011), The dependency of geoengineered sulfate aerosol on the emission strategy. Atmosph. Sci. Lett., 12: 189-194. https://doi.org/10.1002/asl.304
Citation: https://doi.org/10.5194/gmd-2023-79-RC2